# Hilltop curvature as a proxy for erosion rate: Wavelets enable rapid computation and reveal systematic underestimation

William T. Struble[1,2], Joshua J. Roering[2]

[1]Department of Geosciences, University of Arizona, Tucson, Arizona 85721, USA
[2]Department of Earth Sciences, University of Oregon, Eugene, Oregon, 97403, USA

*Correspondence to*: William T. Struble (wtstruble@arizona.edu)

**Abstract.** Estimation of erosion rate is an important component of landscape evolution studies, particularly in settings where transience or spatial variability in uplift or erosion generates diverse landform morphologies. While bedrock rivers are often used to constrain the timing and magnitude of changes in baselevel lowering, hilltop curvature (or convexity), $C_{HT}$, provides

an additional opportunity to map variations in erosion rate given that average slope angle becomes insensitive to erosion rate owing to threshold slope processes. $C_{HT}$ measurement techniques applied in prior studies (e.g. polynomial functions), however, tend to be computationally expensive when they rely on high resolution topographic data such as lidar, limiting the spatial extent of hillslope geomorphic studies to small study regions. Alternative techniques such as spectral tools like continuous wavelet transforms present an opportunity to rapidly document trends in hilltop convexity across expansive areas.

Here, we demonstrate how continuous wavelet transforms (CWTs) can be used to calculate the Laplacian of elevation, which we utilize to estimate erosion rate in three catchments of the Oregon Coast Range that exhibit varying slope angle, slope length, and hilltop convexity, implying differential erosion. We observe that $C_{HT}$ values calculated with the CWT are similar to those obtained from 2D polynomial functions. Consistent with recent studies, we find that erosion rates estimated with $C_{HT}$ from both CWTs and 2D polynomial functions are consistent with erosion rates constrained with cosmogenic

radionuclides from stream sediments. Importantly, our CWT approach calculates curvature at least $10^3$ times more quickly than 2D polynomials. This efficiency advantage of the CWT increases with domain size. As such, continuous wavelet transforms provide a compelling approach to rapidly quantify regional variations in erosion rate as well as lithology, structure, and hillslope sediment transport processes, which are encoded in hillslope morphology. Finally, we test the accuracy of CWT and 2D polynomial techniques by constructing a series of synthetic hillslopes generated by a theoretical

nonlinear transport model that exhibit a range of erosion rates and topographic noise characteristics. Notably, we find that

neither CWTs nor 2D polynomials reproduce the theoretically prescribed $C_{HT}$ value for hillslopes experiencing moderate to fast erosion rates, even when no topographic noise is added. Rather, $C_{HT}$ is systematically underestimated, producing a power law relationship between erosion rate and $C_{HT}$ that can be attributed to the increasing prominence of planar hillslopes that narrow the zone of hilltop convexity as erosion rate increases. As such, we recommend careful consideration of measurement length scale when applying $C_{HT}$ to estimate erosion rate in moderate to fast-eroding landscapes, where curvature measurement techniques may be prone to systematic underestimation.

## 1 Introduction

The morphology of landscapes adjusts to conform to exogenic perturbations such as uplift and climate as well as spatial variations in lithology, geologic structure, and biology. As such, numerous studies have taken advantage of landscape morphology to estimate rates and timing of perturbations to these landscape properties. In bedrock rivers, for instance, geomorphic transport laws have been formulated to allow for linkages between landscape form and process, including from measurements such as channel steepness and $\chi$, a metric that integrates drainage area along a channel profile (Kirby and Whipple, 2001; Perron and Royden, 2013; Royden and Perron, 2013). These tools have been effectively utilized to estimate and map spatial variations in uplift, quantify the timing and rates of landscape transience and uplift history, and predict drainage basin reorganization (e.g. Barnhart et al., 2020; Dietrich et al., 2003; Fox, 2019; Kirby and Whipple, 2001, 2012; Roberts and White, 2010; Willett et al., 2014; Wobus et al., 2006).

Similarly, hillslope geomorphic transport laws formulated for soil mantled landscapes allow for estimation of uplift and erosion rates as well as prediction of the migration of hillcrests in response to landscape transience (Forte and Whipple, 2018; Mohren et al., 2020; Mudd, 2017; Mudd and Furbish, 2007, 2005; Roering, 2008; Roering et al., 2007, 2001, 1999). Over 100 years ago, it was proposed that hillslope form, specifically slope and curvature, may be an effective predictor of erosion rate, as hillslopes steepen and lengthen to accommodate increases in baselevel lowering (Gilbert, 1909, 1877). However, hillslopes do not continue to steepen as baselevel lowering progressively increases to faster and faster rates (e.g. Howard, 1994; Penck, 1953; Schumm, 1967; Strahler, 1950). Rather, hillslope gradients approach a threshold value as erosion rate increases, such that gradient becomes invariant and insensitive to further increases in baselevel lowering (Andrews and Bucknam, 1987; Burbank et al., 1996; DiBiase et al., 2012; Larsen and Montgomery, 2012; Montgomery, 2001; Roering et al., 1999). In such cases, sediment flux varies nonlinearly with slope due to threshold-dependent processes such as landsliding as well as granular creep (BenDror and Goren, 2018; Deshpande et al., 2021; DiBiase et al., 2012; Ferdowsi et al., 2018; Gabet, 2000; Larsen and Montgomery, 2012; Montgomery, 2001; Ouimet et al., 2009; Roering et al., 2001).

Despite the insensitivity of hillslope gradient in rapidly eroding landscapes, soil mantled hillslopes remain an effective record of landscape transience and uplift. Specifically, hilltop curvature continues to respond to baselevel lowering when uplift

and erosion rates are high, even as slope becomes insensitive to ever-increasing erosion rate (Hurst et al., 2012; Mohren et al., 2020; Roering et al., 2007). For a one-dimensional hillslope at steady state, erosion rate, E [L/T], can be estimated as

$$E = -\frac{\rho_s}{\rho_r} D C_{HT} ,$$ (1)

where $\rho_s$ and $\rho_r$ are the density of soil and bedrock [M/L$^3$], respectively, D is the soil transport coefficient or diffusivity [L$^2$/T], and $C_{HT}$ is curvature at the hilltop [1/L] (Roering et al., 2007). Using this formulation, Hurst et al. (2012) demonstrated in the Sierra Nevada, California, that $C_{HT}$ records erosion rate in both low-relief, low-slope headwater catchments of the Feather River as well as in high-relief catchments that have already adjusted to a faster baselevel lowering rate where hillslopes approach a threshold angle. Similarly, Hurst et al. (2013) observed that hillslopes that are translating through an uplift gradient along the San Andreas Fault actively steepen and become sharper ($C_{HT}$ becomes more negative) as they traverse the zone of high uplift and hillslope gradients become invariant. The hillslopes then decay, that is slopes become gentler and curvatures become less sharp, as they reenter the region of low background uplift (Hurst et al., 2013). Similarly, Clubb et al. (2020) observed that steep channels and sharp hilltops record uplift along the Mendocino Triple Junction in northern California, and they note that the lag in hillslope response time relative to the bedrock channels records the northward migration of the Mendocino Triple Junction.

Past studies that couple geomorphic transport laws and hilltop curvature have typically relied on curvature calculated from 2D polynomial functions fit to the topographic surface (PFTs (i.e. polynomials fit to topography); e.g. Roering et al., 1999). While a variety of polynomial forms and types of curvature (i.e. tangential, planform, Laplacian, etc.) have been utilized (e.g. Minár et al., 2020; Moore et al., 1991), Hurst et al. (2012) found that 6 term functions were sufficient for measuring curvature to estimate erosion rate. Specifically, Hurst et al. (2012) used least squared regression to fit a surface, z, to topography, such that,

$$z = ax^2 + by^2 + cxy + dx + ey + f ,$$ (2)

where curvature, or more specifically the Laplacian of elevation, $\nabla^2 z$, is denoted as

$$\nabla^2 z = 2a + 2b.$$ (3)

To reduce the impact of topographic roughness due to stochastic sediment transport and surface perturbations such as boulders and tree throw pits as well as noise in the digital topographic data, they applied the PFT over a scale, $\lambda$ (L in Hurst et al., (2012)), which defines the size of the square polynomial kernel that is fit to the surface. The value of $\lambda$ [L] can be obtained by analysis of the scale dependency of roughness metrics (e.g. Hurst et al., 2012; Roering et al., 2010). As elaborated in the methodology proposed by Hurst et al. (2012), the PFT is not required to pass through any digital elevation model (DEM) nodes; hence, $\lambda$ can be understood as a smoothing scale, thus measuring the background $C_{HT}$ and removing topographic noise.

While the application of PFTs has proven useful for calculating curvature to estimate erosion rate and predict spatial and temporal variations in uplift (e.g. Clubb et al., 2020; Godard et al., 2020; Hurst et al., 2019, 2013, 2012; Mohren et al.,

2020; Roering et al., 2007), PFTs are computationally cumbersome, hindering large-scale exploitation of high-resolution topographic datasets that have become increasingly available. Here, we demonstrate that 2D continuous wavelet transforms (CWTs) provide an alternative and computationally efficient approach to calculating hilltop curvature, operating at least $10^2$ to $>10^3$ times faster than PFTs, with the relative efficiency advantage of CWTs increasing with the size of digital elevation models. We establish the similarity of the output CWT $C_{HT}$ values to those produced by PFTs, and we compare estimated erosion rates calculated from $C_{HT}$ values to erosion rates measured with cosmogenic radionuclides (CRN) in catchments in the Oregon Coast Range. In addition, we test the relative accuracy of the CWT and PFT approaches by applying them to synthetic hillslopes with known erosion rates generated by a nonlinear transport model and superimposed topographic noise. We find that both techniques systematically underestimate $C_{HT}$ at moderate to high erosion rates and appear to approximate a square root relationship between $C_{HT}$ and erosion rate as erosion rate increases, consistent with a recent study (Gabet et al., 2021).

## 2 Study Site: Oregon Coast Range

We selected the Oregon Coast Range (OCR) to compare CWTs and PFTs as hilltop curvature measurement techniques, as it is a region that has been extensively studied in the geomorphic literature, exhibits relatively uniform topography over intra-catchment scales while exhibiting diversity in hillslope form and erosion rate across the axis of the range, and has negligible spatial variability in climate. The OCR is an unglaciated humid landscape that parallels the Cascadia Subduction Zone and is characterized by cool, wet winters when the majority of the annual 1-2 m of precipitation falls, and warm dry summers (PRISM Climate Group, 2016). The dominant tree populations are composed of Douglas-fir (*Pseudotsuga menziesii*) and western hemlock (*Tsuga heterophylla*) that reside on hillslopes that are soil-mantled throughout the range. Soils are thickest in colluvial hollows and unchannelized valleys (~1-2 m), thinnest (~0.5 m) on planar hillslopes and hilltops, and are primarily produced stochastically through tree throw and bioturbation (Dietrich and Dunne, 1978). Colluvial hollows are periodically evacuated by shallow landslides that mobilize into debris flows (Benda and Dunne, 1997; Dietrich and Dunne, 1978; Penserini et al., 2017; Stock and Dietrich, 2003). Erosion rate, measured using techniques including CRNs, $^{14}$C dating, and fluvial and colluvial sediment flux, usually cluster at approximately 0.1 mm yr$^{-1}$ (Balco et al., 2013; Bierman et al., 2001; Heimsath et al., 2001; Penserini et al., 2017; Reneau and Dietrich, 1991), though these rates can temporally and spatially vary dramatically (Almond et al., 2007; Marshall et al., 2015; Sweeney et al., 2012). Average OCR erosion rates approximately correspond with uplift rates calculated from abandoned marine terraces, ranging from <0.05 to >0.4 mm yr$^{-1}$ (Kelsey et al., 1996), as well as from fluvial strath terraces which range from 0.1 to 0.3 mm yr$^{-1}$ (Personius, 1995), which has led to suggestions that the OCR may approximate steady state. Nonetheless, deviations from uniform erosion have been noted based on morphologic trends as well as soil properties (Almond et al., 2007; Sweeney, et al., 2012).

We pinpointed catchments in the OCR that exhibit a range of hilltop curvatures for analysis. Specifically, we focus on Hadsall Creek (43.983° N, -123.823° W), the North Fork Smith River (NFSR; 43.963° N, -123.811° W), and Bear Creek (44.181° N, -123.371° W). Hadsall Creek and the NFSR are catchments in the central OCR that share a drainage divide (Fig.

1A). Hadsall Creek is characterised by steep channels and hillslopes with evenly spaced ridges and valleys where incision is

dominated by debris flows (Penserini et al., 2017; Fig. 2A). Contrastingly, the NFSR, which is erosionally isolated from baselevel by an Oligocene-age gabbroic dike that has pinned the fluvial channel, exhibits comparatively gentle channel and hillslope angles as well as longer soil residence times (Sweeney et al., 2012; Fig. 2B). CRN measurements have recorded catchment-averaged erosion rates at Hadsall Creek and the NFSR of $0.113\pm0.018$ mm yr$^{-1}$ and $0.058\pm0.0054$ mm yr$^{-1}$, respectively (recalculated from Penserini et al., 2017; Table 1). We also utilize hillslopes within three small sub-catchments

that drain to Bear Creek (Fig. 1B), a tributary to the Long Tom River on the eastern margin of the OCR in the southwestern Willamette Valley (WV). Hillslopes within Bear Creek and the western margin of the WV exhibit gentle slopes, weathered soils with long residence times >150 kyr (Almond et al., 2007), and are bounded by broad alluviated valleys (Fig. 2C). We additionally report a newly collected CRN-derived catchment-averaged erosion rate for the northern Bear Creek subcatchment that we study here (Fig. 1B).

The spatial proximity of Bear Creek, Hadsall Creek, and the NFSR make them well-suited to compare $C_{HT}$ measurement techniques, as other factors that may influence morphology, such as climate and lithology, remain relatively invariant. All three catchments are within the Tyee Formation, a ~3 km thick sequence of gently dipping Eocene turbidite deposits characterized by a sequence of sandstone and siltstone interbeds (Baldwin, 1956; Heller and Dickinson, 1985; Lovell, 1969). While variability in sandstone-siltstone ratios in the Tyee Formation result in latitudinal north-south variations in deep-

seated landsliding (Roering et al., 2005), our three study sites are within sufficient proximity to each other such that lithologic variability in setting hillslope morphology should be limited. In addition, while common elsewhere in the OCR (Franczyk et al., 2019; LaHusen et al., 2020; Roering et al., 2005), the sites we have selected for analysis do not exhibit pronounced evidence of deep-seated landslides, which may bias $C_{HT}$ values, complicating comparison to known erosion rates from CRN analysis. As such, Hadsall Creek, the NFSR, and Bear Creek provide an ideal spectrum of hillslopes that allows for assessment of $C_{HT}$

measurement techniques (Fig. 2).

## 3 Methods

### 3.1 Curvature calculation: polynomial fit and continuous wavelet transform

We used PFTs to calculate curvature of the Hadsall and Bear Creeks and NFSR lidar DEMs as enumerated in Equations 2 and 3. Each DEM has a grid spacing of 0.9144 m (3 ft). The lidar for Bear Creek was collected in 2009 (average

point density: 8.14 pulses m$^{-2}$, ground density: 1.36 pulses m$^{-2}$), and the lidar at Hadsall Creek and NFSR was collected in 2014 (average point density: 10.41 pulses m$^{-2}$, ground density 0.54 pulses m$^{-2}$). (See Code and Data Availability for access information to lidar data). In order to identify and remove the topographic impact of stochastic sediment transport processes such as tree throw, we calculated PFT curvature rasters using variable kernel sizes, corresponding to a range of smoothing scales, specifically for $\lambda$=5-141 m (the diameter of the polynomial kernel requires odd dimensions). These values of $\lambda$ are

informed by the requirements of the CWT, which we discuss below. For the PFT, $\lambda$ is the diameter of the smoothing window.

In contrast to PFTs, CWTs are computationally efficient and can provide a variety of outputs depending on the analysis and type of wavelet used (e.g. Foufoula-Georgiou and Kumar, 1994 and references therein). Here, we applied a 2D CWT using the Ricker wavelet (often known as the Mexican Hat wavelet). The Ricker wavelet has been used in geomorphology to map and estimate landslide ages based on surface roughness (Booth et al., 2009; LaHusen et al., 2020),

identify dominant landforms at particular wavelengths (Struble et al., 2021), extract channel heads and drainage networks (Lashermes et al., 2007; Passalacqua et al., 2010), and other topographic spectral analyses including mapping faults and predicting lithospheric thickness (e.g. Audet, 2014; Jordan and Schott, 2005; Malamud and Turcotte, 2001). In applying the Ricker wavelet, we take advantage of a useful property of convolutions that allows for simultaneous removal of topographic noise and calculation of derivatives. Specifically,

$\frac{\partial}{\partial x}(f * h) = \frac{\partial f}{\partial x} * h = f * \frac{\partial h}{\partial x}$, (4)

where $f$ is some function (topography in our case) and $h$ is a smoothing function (2D Gaussian for instance). Hence, for the case of calculating derivatives of topography, Equation 4 implies that applying a low-pass filter to topography and then taking the derivative (left term of Equation 4) is identical to taking the derivative of topography and smoothing the outputs (middle term), which is correspondingly equivalent to taking the derivative of the smoothing function and using that function

to smooth topography (right term; Lashermes et al., 2007). The application of the Ricker wavelet to calculate land surface curvature is akin to the rightmost term in Equation 4, albeit by taking the second derivative of the smoothing function.

The Ricker wavelet is the negative second derivative of a 2D Gaussian function [$1/L^2$], which is given as

$g(x,y) = \frac{1}{2\pi s^2} \exp\left[-\frac{(u-x)^2 + (v-y)^2}{2s^2}\right]$, (5)

where $(u,v)$ and $s$ [L] ($\sigma$ in Lashermes et al., 2007) define the location and size, specifically the standard deviation, of the

Gaussian function, respectively (Derivative of a Gaussian (DoG) wavelets constitute a wavelet family). The Ricker wavelet, $\psi$ [$1/L^4$], as the negative, second derivative of Equation 5, then, is defined as

$\psi(x,y) = \frac{1}{\pi s^4}\left(1 - \frac{1}{2}\left(\frac{(u-x)^2 + (v-y)^2}{s^2}\right)\right) \exp\left[-\frac{((u-x)^2 + (v-y)^2)}{2s^2}\right]$. (6)

The generalized 2D CWT of topography, $z$, at location $(u,v)$, then is given as

$C(s,u,v) = \frac{1}{s}\int_{-\infty}^{\infty}\int_{-\infty}^{\infty} z(x,y)\psi\left(\frac{x-u}{s},\frac{y-v}{s}\right)dxdy$, (7)

Equation 7, notably, is a convolution of $z$ and $\psi$, expressed as

$C(s,u,v) = z(x,y) * \psi\left(\frac{x-u}{s},\frac{y-v}{s}\right)$, (8)

where ∗ represents the convolution. The output wavelet coefficients, *C*, of the Ricker wavelet from Equation 8 provide a measure of the low-pass filtered Laplacian over the input scale of interest (Foufoula-Georgiou and Kumar, 1994; Lashermes et al., 2007) that we use to estimate curvature of extracted hilltops ($C_{HT}$).

Similar to the application of PFTs to estimate erosion rate, it is necessary to select a measurement scale that effectively smooths over stochastic sediment transport perturbations and noise that is inherent to topographic datasets and DEMs and does not represent long-term morphology reflective of baselevel lowering (Hurst et al., 2012; Roering et al., 2010). Thus, it is important to utilize an appropriately scaled wavelet, *s* (akin to a kernel size), to generate curvature values that are appropriate to represent $C_{HT}$. Several definitions for the smoothing scale of a DoG wavelet exist. Torrence and Compo (1998) define the

smoothing scale, λ [L], for an m$^{th}$ DoG as

$$\lambda = \frac{2\pi s}{\sqrt{m+\frac{1}{2}}} \, . \tag{9}$$

For the Ricker wavelet, m=2. Under the Torrence and Compo (1998) definition, λ is determined by the scale, *s*, at which the wavelet power spectrum applied for a particular function with a known frequency (i.e. a cosine function) attains a maximum value. Alternatively, Lashermes et al. (2007) define the Ricker wavelet smoothing scale as the inverse of the wavelet's band-

pass frequency, such that

$$\lambda = \sqrt{2}\pi s \, . \tag{10}$$

To clarify, while λ represents the physical scale at which topography is smoothed, *s* specifically defines the scale of the wavelet function and is related to the physical smoothing scale through Equations 9 and 10 and is not interchangeable with λ. While the Torrence and Compo (1998; TC98) and Lashermes et al. (2007; L07) λ definitions generate similar smoothing scales, the

output Laplacian values may be sufficiently diverse to produce significantly different erosion rate estimates depending on the choice. Thus, we utilize both definitions by selecting a range of λ and solving for *s* using both Equations 9 and 10 in order to apply the CWT, which we then compare to the curvature values produced from the PFT.

      We applied the CWT and PFT for λ values that correspond to the scales at which topographic noise manifests in topographic data. The CWT can only be applied for s>1, which for DEMs with a grid spacing of ~1 m with the odd-dimensions

constraint of the PFT, places a lower λ limit of 5 m. We additionally tested larger λ (up to 141 m) to isolate the consistency between the CWT and PFT. For each smoothing scale, λ, for which we calculated curvature, we solved for s in Equations 9 and 10 to construct the appropriately sized Ricker wavelet (Equation 6). We then applied the CWT to the OCR lidar DEMs for smoothing scales of 5-141 m (same as PFT) and produced $C_{HT}$ values for the CWT and PFT methods, denoted as $C_{HT-W}$ and $C_{HT-P}$, respectively.

### 3.2 Computational efficiency of curvature values

We compared the efficiency of calculating curvature with a PFT to the CWT, including both definitions of wavelet smoothing scale, $\lambda$ (TC98 and L07; Equations 9, 10). We measured curvature for $\lambda=5\text{-}197$ m in MATLAB on a personal laptop with 16 GB of RAM (2.60 GHz CPU). To account for potential variations in calculation time that may result from variable landscape morphology, we utilized sample regions of the Hadsall Creek and Bear Creek DEMs, as they represent the high and low erosion rate end members of our test sites. Each DEM was a 513x513 single precision grid (32-bit float) with a cell size of 0.9144 m.

We also tested how DEM size affects the relative speed of the CWT and PFT algorithms. We selected a DEM of size 682x682 pixels from the Hadsall Creek catchment and measured curvature for $\lambda=5\text{-}101$ m. We then calculated curvature for the same $\lambda$ on the northwest quadrant of the 682x682 pixel DEM, corresponding to a 341x341 pixel medium-sized grid. Finally, we calculated curvature for $\lambda=5\text{-}101$ m on the northwest quadrant of the medium-sized DEM, corresponding to a 171x171 pixel grid.

### 3.3 Hilltop extraction

We calculated curvature at every pixel of our DEMs, but $C_{HT}$ requires limiting curvature values to hilltop pixels. Therefore, we extracted hilltop masks in MATLAB with the DIVIDEobj function of TopoToolbox (Scherler and Schwanghart, 2014), restricting extracted first-order divides to those with lengths exceeding 800 m (Schwanghart and Scherler, 2020). We further refined the hilltop masks by only considering locations where $C_{HT}$ is negative (convex) and where local hillslope gradient is less than 0.4, above which a greater proportion of hillslope sediment transport can be classified as nonlinear. We manually removed drainage divides mapped in low-relief valley bottoms and where flow routing is interrupted by roads, which are common in the OCR and introduce noisy high-magnitude curvatures. While the signature of deep-seated landslides is generally absent from our study catchments, if it appeared in the DEM that there has been a history of bedrock slope instability, we filtered hilltops proximal to mapped landslides. We also did not consider hilltops that may exhibit prominent asymmetry due to disequilibrium with neighboring drainage basins. Thus, at Hadsall Creek and NFSR, we neglected all hilltops at the main drainage divide (Fig. 1A). At the Bear Creek catchments, we similarly removed all hilltops at the main drainage divide (the northeast divide in Fig. 1B) except for those that border adjacent catchments that are likely experiencing the same baselevel imposed by Bear Creek (southwest divide in Fig. 1B). Finally, to visualize the scale-dependency of $C_{HT}$ and reduce potential noise in $C_{HT}$ measurements for full catchments obscuring curvature scaling breaks (Hurst et al., 2012; Roering et al., 2010), we selected a single representative hilltop in each catchment (Fig. 1, 2), chosen such that it approximates the average curvature for the catchment when compared to curvature measurements taken for all hilltops (Fig. 3). The selected representative hilltop spans 234 m at Hadsall Creek (average gradient 0.23), 149 m at NFSR (average gradient 0.14), and 274 m at Bear Creek (average gradient 0.11).

### 3.4.1 Erosion rates estimated from hilltop curvature

We applied the CWT and PFT to the Hadsall Creek, NFSR, and Bear Creek lidar DEMs, and calculated curvature. We utilized the hilltop masks to extract curvature at the hilltops ($C_{HT}$). In the OCR, Roering et al. (2010) observed a scaling break in curvature at 15 m, corresponding to the length scale below which pit and mound topography dominate the surface morphology. We observe similar scaling breaks in hilltop curvature for selected hilltops at $\lambda \approx 15\text{-}20$ m (Fig. 3), though we note that the clarity of this scaling break depends on the size of the study area and consistency, or lack thereof, of small pit and mound topography in a landscape. Thus, while the scaling break that distinguishes the effective scale at which topographic noise is filtered out may differ between the DEMs and catchments we analyse here, we find that the scaling breaks do not clearly or systematically differ from those observed by Hurst et al. (2012) and Roering et al. (2010; Fig. 3). Thus, we used a smoothing scale of $\lambda = 15$ m for the PFT and CWT in each OCR catchment to estimate erosion rate as enumerated in Equation 1. We assumed that $\frac{\rho_s}{\rho_r} = 0.5$ and D=0.003 m$^2$ yr$^{-1}$, a hillslope diffusivity estimated for the OCR (Roering et al., 1999, 2007). We compared the mean and variance of these estimated erosion rates to CRN-derived erosion rates in each OCR study catchment.

### 3.4.2 Erosion rates from cosmogenic radionuclides

To test the efficacy of $C_{HT}$ as a proxy for erosion rate, we compare erosion rates estimated from $C_{HT}$ to those estimated from CRNs in stream sediments. We collected stream sediments from the western tributary to Bear Creek that we study here (Fig. 1B; 44.186 °N, -123.375° W) to estimate erosion rate with cosmogenic $^{10}$Be (Balco et al., 2013; Heimsath et al., 2001). We used the online calculator CRONUS (Balco et al., 2008) to calculate erosion rate for the sample, which incorporates the material from the upstream drainage area and assumes steady erosion over the CRN integration timescale (Table 3, 4). We additionally recalculate the erosion rates for Hadsall Creek and NFSR from the CRN data previously reported by Penserini et al. (2017; Table 2, 3).

### 3.5 Construction of synthetic hillslopes to test $C_{HT}$ measurements

We utilized synthetic hillslopes generated from a theoretical model to compare the accuracy of hilltop curvature calculated using the PFT and CWT as well as test how well these approaches can predict erosion rate. We used the functional form for a 1D hillslope experiencing nonlinear diffusion given as

$$z = \frac{DS_c^2}{2(\rho_r/\rho_s)E}\left[\ln\left(\frac{1}{2}\left(\sqrt{1+\left(\frac{2\left(\frac{\rho_r}{\rho_s}\right)Ex}{DS_c}\right)^2}+1\right)\right)-\sqrt{1+\left(\frac{2\left(\frac{\rho_r}{\rho_s}\right)Ex}{DS_c}\right)^2}+1\right], \tag{11}$$

where E is the erosion rate calculated using Equation 1 [L/T], $S_c$ is the threshold, or critical, slope angle, and x is distance along the hillslope profile (Roering et al., 2007). We extended the hillslope profile solution perpendicular to the x-axis to construct a 2D synthetic hillslope on a 201x201 m grid (Fig. 8, 9, S2). Odd hillslope dimensions ensure the existence of a hilltop pixel in the middle of the domain. We utilized the PFT and CWT, including both CWT definitions for the wavelet scale

$\lambda$ (Equations 9, 10; TC98, L07), to calculate $C_{HT-W}$ and $C_{HT-P}$ of the synthetic hillslopes for several different scenarios. Specifically, we considered various dimensionless erosion rates, E*, given by:

$$E^* = \frac{2E\left(\frac{\rho_r}{\rho_s}\right)L_H}{DS_c} = \frac{2C_{HT}L_H}{S_c}, \tag{12}$$

where $L_H$ is hillslope length (Roering et al., 2007). In testing the ability of the CWT and PFT to predict hilltop curvature, we generate hillslopes with a range of E* values that can account for variations in E, $C_{HT}$, $L_H$, and $S_c$. For instance, low (high) E* values may correspond to low (high) E, $C_{HT}$, or $L_H$ as well as high (low) $S_c$, or some combination thereof. We specifically tested E* values of 1, 10, 30, and 100. While E*=100 is an extreme case and may only be rarely observed in natural landscapes that are eroding rapidly and also manage to maintain a soil mantle, such as badlands, E* values of 1, 10, and 30 have been readily observed in multiple landscapes. For instance, Hurst et al. (2012) observed E* values in the Sierra Nevada, California, as high 30, but most measured hillslopes exhibited E*<10. Hurst et al. (2013) similarly observed at the Dragon's Back along the San Andreas Fault that E* ranges from 5 to 30, reflecting a strong uplift gradient. E* along coastal California has been noted as <~10 along the Mendocino Triple Junction (Clubb et al., 2020), <~8 further south along the Bolinas Ridge (Hurst et al., 2019), and ranging from 1-2 at Gabilan Mesa (Roering et al., 2007). Finally, in the OCR, past studies have observed that E*≈10 (Marshall and Roering, 2014; Roering et al., 2007).

In addition, to account for natural topographic roughness that the CWT and PFT smooth over to estimate $C_{HT}$, we introduce noise to the synthetic hillslopes in the form of white ($\beta$=0), pink ($\beta$=-1), and red, or Brownian, ($\beta$=-2) noise, where $\beta$ is spectral slope. White noise denotes a random surface where all wavenumbers (frequencies) have equal amplitude, or spectral power. Conversely, spectral power density varies inversely ($\beta$=-1) with wavenumber for pink noise, such that low wavenumbers have higher intensity. Similarly, red noise exhibits higher spectral power at low wavenumbers, but more dramatically than for pink noise. While hillslope spectra will vary between landscapes and likely exhibit a combination of different spectral slopes depending on the scale of analysis, red noise surfaces generally best describe topographic noise in natural landscapes while white noise surfaces are comparatively the least likely (e.g. Booth et al., 2009; García-Serrana et al., 2018; Marshall and Roering, 2014; Pelletier and Field, 2016; Perron et al., 2008). We generated each noisy surface of values normally distributed about 0 with the standard deviation ranging from -1 m (pits) to 1 m (mounds; Konowalczyk, 2021). For each type of noise, we tested how the amplitude of the noise affects calculated $C_{HT}$ by scaling the noise distributions (±1 m) by 0.1%, 0.5%, and 5% of hillslope length ($L_H$=100 m). In other words, we test cases where the standard deviation of the noise, $\sigma$, is $\sigma$=0.001$L_H$, $\sigma$=0.005$L_H$, and $\sigma$=0.05$L_H$, corresponding to 1$\sigma$ values of 10 cm, 50 cm, and 5 m, respectively. While topographic noise with a distribution of amplitudes with a standard deviation of 5 m is likely unphysical for soil mantled landscapes, this extreme case allows us to clearly test how different topographic parameters affect calculated values of E* and how well each measurement technique can filter out noise.

## 4 Results

### 4.1 Computational efficiency of CWT and PFT curvature calculation

We find that the CWT is dramatically more efficient at calculating hilltop curvature than the PFT. Curvature
calculation time depends on smoothing scale, $\lambda$, with large kernel sizes taking longest for both the PFT and CWT. Specifically,
we compared curvature calculation times for the PFT and CWT in selected portions of the Hadsall Creek and Bear Creek
catchments for $\lambda$=5-197 m. We find that for the 513x513 single precision grid, the PFT takes ~4-4.5 seconds to calculate
curvature at the smallest scales and ~30 seconds to calculate curvature at larger scales. Measurement time does not vary greatly
between the fast and slowly eroding landscape DEMs. By comparison, for $\lambda$=5-197 m, both the CWT L07 and TC98 definitions
for $\lambda$ calculate curvature at the smallest scales in ~0.0039-0.004 seconds while at larger scales they calculate curvature in ~2.2-
2.3 seconds. Comparing the two techniques, we find that at the smallest smoothing scales ($\lambda$=5m) the CWT operates >$10^3$
times faster than the PFT, while at larger scales where $\lambda$ approaches 200 m, the CWT still outpaces the PFT by over an order
of magnitude (Fig 4A, B).

In addition to the CWT outpacing the PFT at a large range of $\lambda$ in two landscapes exhibiting contrasting morphology,
we observe that the relative speed of the CWT to the PFT increases with DEM size. Specifically, we find that for the smallest
DEM for which we calculated curvature (171x171 grid), the CWT is ~500 times faster than the PFT when $\lambda$=5 m and is ~10
times faster than the PFT when $\lambda$=101 m (Fig. 4C). As DEM size increases, the computational advantage of the CWT increases,
such that for the large DEM (682x692 grid), the CWT operates >$10^3$ times faster than the PFT when $\lambda$=5 m and ~30 times
faster when $\lambda$=101 m (Fig. 4C).

## 4.2 Similarity of $C_{HT-P}$ and $C_{HT-W}$

We utilized 2D CWTs and PFTs to calculate $C_{HT-W}$ and $C_{HT-P}$ for a range of $\lambda$ in the OCR catchments of Hadsall
Creek, NFSR, and Bear Creek. We find that $C_{HT-W}$ and $C_{HT-P}$ are similar when using $\lambda$ values of 5-30 m. Specifically, Fig. 3
compares output $C_{HT-W}$ using both $\lambda$ length scale definitions (Equations 9, 10) and $C_{HT-P}$ for the representative hilltop in each
catchment. Mean measured $C_{HT-W}$ and $C_{HT-P}$ values differ the most at small smoothing scales, where signal to noise ratio
(topographic noise to underlying $C_{HT}$) is highest (Fig. 3A, D, G). At these small smoothing scales, the standard deviation of
$C_{HT-P}$ is larger than that of $C_{HT-W}$ (Fig. 3B, E, H). Mean $C_{HT-W}$ for both L07 and TC98 $\lambda$ definitions are similar, as are the output
standard deviations (Fig. 3). However, we observe that mean $C_{HT-W}$ calculated using the TC98 definition of $\lambda$ is lower in
magnitude than that of L07 (Fig. 3; Table 1). This is not unexpected, however, since $\lambda$, as defined by TC98 in Equation 9, is
effectively smaller than that of L07 defined in Equation 10, for a given wavelet scale, $s$. Figure 4 compares the output $C_{HT}$
measurements from each technique by plotting $C_{HT}$ for individual DEM nodes for $\lambda$=15 m. If measurements from each
technique are in agreement, their output $C_{HT}$ values should plot as a 1:1 line. Indeed, $C_{HT-W}$ for TC98 $\lambda$ is lower than that of
L07 for both the representative hilltop and all mapped hilltops, with the largest deviation occurring on the sharpest hilltops
(Fig. 5C, F, I). Similarly, mean $C_{HT-W}$ (TC98 and L07) is lower than $C_{HT-P}$, particularly for high magnitude curvatures.
Nevertheless, the output values from each definition do not vary dramatically, particularly when considering the $C_{HT}$ for DEM
nodes corresponding to representative hilltops (Fig. 5).

We additionally plot probability density functions (PDF) of measured $C_{HT-W}$ and $C_{HT-P}$ for each catchment (Fig. 6, S1). Notably, the shape of each PDF is similar between measurement techniques but is shifted along the x-axis due to the variable definitions of λ. This shift is further illustration of the deviation from a 1:1 relationship between each measurement

technique as observed in Fig. 5. Similar to the greater deviation between calculated $C_{HT}$ at curvature extrema in Fig. 5, we observe greater offset between PDFs in the distribution tails, while the peaks remain similar. We observe this consistency between PDF peaks reflected in the mean $C_{HT}$ of the PDFs, which are similar regardless of measurement technique (Fig. 6; Table 1, 2).

## 4.3 Erosion rate calculated with $C_{HT}$ and cosmogenic radionuclides

We utilized $C_{HT}$ for λ=15 m to estimate erosion rate. Erosion rates calculated from $C_{HT-P}$ and $C_{HT-W}$ for all mapped hilltops and the representative hilltop in each catchment can be found in Table 2. We observe that $C_{HT-P}$ and $C_{HT-W}$ produce expected relative pattern of erosion rate in our OCR catchments. That is, calculated erosion rate from $C_{HT}$ is fastest at Hadsall Creek and slowest at Bear Creek, as revealed by our cosmogenic erosion rate data (Fig. 6, Table 2). Notably, we observe that $C_{HT}$-generated erosion rates (mean ± standard deviation) fall within or near the measurement uncertainty of the CRN erosion

rate for both the representative hilltop and all hilltops (Table 2). For instance, CRN-measured erosion rates are 0.113±0.018 mm yr$^{-1}$ at Hadsall Creek, 0.058±0.0054 mm yr$^{-1}$ for the NFSR, and 0.008±0.0007 mm yr$^{-1}$ at Bear Creek (Table 3, 4). Similarly, for the case of the representative hilltop and using the TC98 λ definition, we find $C_{HT}$-calculated erosion rates of 0.178±0.030 mm yr$^{-1}$ at Hadsall Creek, 0.088±0.025 mm yr$^{-1}$ for the NFSR, and 0.007±0.005 mm yr$^{-1}$ at Bear Creek. The rates calculated with the PFT and L07 λ definition are similar, whether considering the representative hilltop or all mapped hilltops

in each catchment (Table 2; Fig. 6, 7, S1). Finally, we observe linear correlation between $C_{HT}$-calculated and CRN-measured erosion rates at our OCR catchments (E=0.88$C_{HT}$+0.002), consistent with the relationship between $C_{HT}$ and E expected in Equation 1 (Fig. 7). Furthermore, the diffusivity we infer from the slope of this relationship is 0.002±0.0004 m$^2$ yr$^{-1}$ (taking into account $\frac{\rho_s}{\rho_r} = 0.5$), a value consistent with diffusivities measured elsewhere in the OCR (Roering et al., 1999).

## 4.4 Testing of $C_{HT}$ extraction with synthetic hillslopes

We calculated $C_{HT-P}$ and $C_{HT-W}$ for a series of synthetic hillslopes with a range of dimensionless erosion rates, E*, and topographic noise (Fig. 8, 9). We observe that the ability of the CWT and PFT to reproduce the defined curvature at particular λ depends on the dimensionless erosion rate, E*, though the type and magnitude of added noise contributes to uncertainty in appropriate λ values to be used to calculate erosion rate. We focus on synthetic hillslopes where no noise has been added (i.e.

σ=0 cm) as well as where noise amplitude σ=0.5% $L_H$, as the magnitude of noise in this case (σ=50 cm) is a reasonable physical approximation of noise and surface roughness in natural landscapes (e.g. Marshall and Roering, 2014; Pelletier and Field, 2016; Roth et al., 2020). The cases where noise amplitude is defined by σ=0.1% $L_H$ (σ=10 cm) and σ=5% $L_H$ (σ=5 m) can be found in the Supplemental Information (Fig. S5-S10).

### 4.4.1 Slowly eroding synthetic hillslopes, E*=1

We observe that for E*=1, both the PFT and CWT reasonably predict the model-defined $C_{HT}$ (and thus E) at moderate smoothing scales. Specifically, when σ=0.5% $L_H$, $C_{HT-W}$ and $C_{HT-P}$ converge on the defined $C_{HT}$ when λ>~9-11 m for white noise, λ>~15-19 m for pink noise, and λ>~13 m for red noise (Fig. 10B-D). At smaller λ, the signal to noise ratio is too high for noise be adequately filtered by either the PFT or CWT. This mirrors past results in natural landscapes, where a sufficiently large smoothing scale must be selected to smooth over topographic noise and recover an accurate $C_{HT}$ (Hurst et al., 2012; Roering et al., 2007). Notably, when E*=1, the hillslopes are not sufficiently steep to approach Sc (Fig. 8, 9). Thus, even at the largest smoothing scales, the CWT and PFT accurately record curvature (Fig. 10A-D). In natural landscapes, however, valley bottoms will introduce positive curvatures, which will cause an increase in curvature (i.e. become less negative), at smoothing scales that approach the hillslope length, which has also been utilized to constrain an optimal smoothing scale (Hurst et al., 2012), and which we observe in OCR catchments (Fig. 3A, D, G).

We observe that the uncertainty in $C_{HT}$, which we define as the standard deviation of $C_{HT}$ along the hilltop, is highest at the smallest smoothing scales (Fig. S3, S4). Notably, we observe for all noise types that at small smoothing scales of λ=5-~13m, $C_{HT-P}$ exhibits higher uncertainty than $C_{HT-W}$. As λ increases, the uncertainty in $C_{HT-P}$ and $C_{HT-W}$ diminishes as topographic noise is progressively filtered. Because red noise includes higher spectral power at long wavelengths, we observe that the decrease in $C_{HT}$ uncertainty occurs at larger smoothing scales, converging towards 0 at scales of >17 m (Fig. S4).

We find that when no noise is added to the synthetic hillslopes, $C_{HT-P}$ and $C_{HT-W}$ accurately predict $C_{HT}$ at all scales (Fig. 10A). While there is some deviation between measured and defined $C_{HT}$ at larger scales, this deviation is exceptionally small (<0.5%) and is primarily a result of edge effects that may not be fully clipped for both the CWT and PFT at the edge of the synthetic hillslope domain. Uncertainty in $C_{HT-W}$ and $C_{HT-P}$ is near 0 when no surface noise is added, with deviations again primarily due to the presence of edge effects that are not fully clipped off at the hillslope tips (Fig. S4). Finally, we observe that for a given style and amplitude of added topographic noise, the uncertainty in $C_{HT}$ does not vary with changes in E* (Fig. S4). We do not vary topographic noise as a function of E*, so equal uncertainty over a range of E* values indicates that variable hillslope form as defined by E* does not affect the uncertainty in $C_{HT}$ *along* the hilltop. Given the convolutional form of the CWT in Equation 8 and the distributive property of convolutions, given as

$$f * (h + k) = (f * h) + (f * k), \tag{13}$$

where *f* is the wavelet, *h* is the synthetic hillslope, and *k* is surface noise, the standard deviation of $C_{HT}$ remaining constant as a function of E* is not unexpected.

### 4.4.2 Moderate to fast eroding synthetic hillslopes, E*≥10

We observe that both the CWT and PFT produce biased $C_{HT}$ as E* increases. The deviation between the model-defined and measured $C_{HT}$ progressively grows for larger E*. Specifically, for the case of λ=15 m, when E*=10, we find that $C_{HT-W}$ and $C_{HT-P}$ are within ~10% of the defined $C_{HT}$, with modest dependencies on the type of topographic noise (Fig. 10F, G, H). However, $C_{HT-P}$ and $C_{HT-W}$ are underestimated by >20% for E*=30 hillslopes and by 60% for E*=100 slopes when λ=15

m. This deviation occurs for hillslopes constructed with topographic noise of all types as well as the synthetic hillslopes without added noise (Fig. 10I-L). Even for small $\lambda$, we observe that $C_{HT}$ is systematically underestimated. For the case of E*=30, we observe that $C_{HT-W}$ and $C_{HT-P}$ deviate by 10-25% for $\lambda<15$ m, with the smallest $\lambda$ ($\sim$5-7 m) exhibiting the least deviation, with $C_{HT-P}$ and $C_{HT-W}$ falling within $\sim$10% of the known $C_{HT}$. $C_{HT}$ is reasonably recovered at $\lambda=5$ m for the red noise E*=30 hillslope, despite the noise dominating $C_{HT-W}$ and $C_{HT-P}$ when $\lambda=5$ m for the E*=1, 10 hillslopes. Given the added noise is constant between E* values, this accurate recovery of $C_{HT}$ for E*=30 when $\lambda=5$ m may indicate that planar hillslopes introduce curvature values sufficiently near-zero to cancel out the positive (concave) noise. For $\lambda>15$ m, we observe that $C_{HT}$ is underestimated by at least 25% for all E*=30 hillslopes and >60% for E*=100 hillslopes. As $\lambda$ increases, this deviation systematically grows such that when $\lambda=35$, $C_{HT}$ is underestimated by half for E*=30 hillslopes and $\sim$80% for exceptionally narrow hillslopes where E*=100 (Fig. 10I-P). Importantly, we observe these major deviations for the hillslopes with no added noise as well, indicating that topographic noise is not solely responsible for biased $C_{HT}$.

## 5 Discussion

Application of CWTs and PFTs to measure $C_{HT}$ and estimate erosion rate in soil mantled landscapes such as the OCR produces erosion rate values that are in agreement with those collected from CRNs in stream sediments, though with dramatically disparate efficiencies. Yet, we also observe that while both techniques accurately reproduce hillslope morphology in synthetic landscapes experiencing modest dimensionless erosion rates, both techniques exhibit systematic bias where dimensionless erosion rate is moderate to high, calling into question the accuracy of past estimates of erosion rate in landscapes that are experiencing moderate to rapid erosion rates. Nevertheless, CWTs are an exciting tool to be added to hillslope geomorphometric analyses, particularly as high-resolution topographic datasets continue to grow and classification of topographic roughness, particularly on the hillslope scale, continues to improve.

### 5.1 $C_{HT}$ measurement and erosion rate estimation in natural landscapes: Oregon Coast Range

We utilized CWTs and PFTs to estimate erosion rate in a landscape that has been thoroughly studied in past geomorphology studies. Encouragingly, $C_{HT}$-calculated erosion rate in Hadsall Creek, NFSR, and Bear Creek reproduce CRN-measured erosion rates from each site. We also observe, however, that some variability in measured $C_{HT}$ reinforces the need to use caution when selecting hilltops at which curvature will be extracted, especially in landscapes where topographic noise, including from anthropogenic sources such as roads, as well as landslides and variable lithology, may introduce inaccurate measurements of curvature. Indeed, despite careful selection of hilltops, calculated $C_{HT}$ exhibit a wide range of values (Fig. 5, 6, S1). Fortunately, the catchments we have sampled here exhibit few to no deep-seated landslides and are mapped entirely within the Tyee Formation, which exhibits little variability over small spatial scales. Also, while there are numerous forest and logging roads throughout the OCR, they are easily identifiable in lidar data and are limited to a small portion of hilltops. Hence, while haphazard selection of hilltops without a predefined methodology for trimming hilltops should be avoided, our observed agreement between estimated erosion rates for all selected hilltops in a catchment and representative hilltops emphasizes that

mild to moderate trimming of hilltop masks is sufficient for estimating an accurate erosion rate (Table 2, Fig. 6, S2). Finally, agreement between TC98 and L07 $\lambda$ definitions and CRN erosion rates suggests that either definition is reasonable for calculating $C_{HT}$. However, careful and informed selection of $\lambda$ when calculating erosion rate remains paramount.

## 5.2 Rapid calculation of $C_{HT}$

We have demonstrated that CWTs calculate $C_{HT}$ $>10^3$ times faster than PFTs at smoothing scales of $\lambda=5$ m (for a 513x513 single precision grid). At smoothing scales often utilized to estimate $C_{HT}$ (~10-30 m), the CWT operates $>10^2$ times faster (Fig. 4). Even at the largest smoothing scales we test (up to 197 m), the CWT operates ~14-15 times faster than the PFT. Importantly, the computational advantage of the CWT increases with DEM size (Fig. 4C), such that the ~$10^3$ computation time advantage that we observe should be considered a minimum, as most landscape analyses utilize DEMs larger than the grids we test here. While PFT computation times can often be substantially reduced by limiting curvature calculation to the hilltops, this dramatic difference in curvature calculation time opens many doors for utilizing hilltop curvature in topographic analyses of landscapes that require consideration of large spatial scales. What's more, the ability of the CWT to operate so efficiently on high-resolution lidar data does not necessitate that coarse data be used to analyse large regions, as has generally been the case for past geomorphic analyses of regional and continental-scale bedrock rivers. Rather, the ability of the CWT to calculate hilltop curvature over large spatial scales with such speed means that the limiting factor for large landscape analyses where lidar data is available is not the operating time of the measurement technique, but rather the ability of existing systems to store vast quantities of high-resolution topographic data and curvature-related products! In addition to $C_{HT}$ measurement, the rapidity of the CWT will allow for large-scale analyses of other types of curvature and landscape characteristics well-suited to spectral analyses including mapping landslides (Booth et al., 2009; LaHusen et al., 2020), quantifying surface roughness (Doane et al., 2019; Roth et al., 2020), and mapping landforms (Black et al., 2017; Clubb et al., 2014; Passalacqua et al., 2010; Perron et al., 2008; Struble et al., 2021).

## 5.3 $C_{HT}$ underestimated in moderate to fast-eroding landscapes

We find that both the CWT and PFT are unable to reproduce accurate $C_{HT}$ at moderate to fast dimensionless erosion rates. Disagreement between measured and defined $C_{HT}$ for a given E* can be conceptualized primarily as a biasing of curvature measurement as hilltops progressively narrow and steepen in response to faster erosion rates. Specifically, since we utilize a nonlinear diffusion framework to construct the synthetic hillslopes (Equation 11), planar side slopes begin to develop and advance towards the hilltop as the hillslope gradient approaches the critical slope angle, $S_c$, at moderate to fast E*. The formation of planar hillslopes means, by definition, that curvature does not accurately reflect $C_{HT}$ along the entire hillslope length, as would be the case for a slowly eroding broad hillslope with constant curvature (i.e. linear diffusion). The E*=1 synthetic hillslope, while also constructed with Equation 11, can be approximated as experiencing linear diffusion, as slopes are not sufficiently steep to approach $S_C$ and develop planarity. In this case, even as $\lambda$ increases, $C_{HT-W}$ and $C_{HT-P}$ accurately

recover the actual $C_{HT}$. We observe that at these slow erosion rates ($E^*$=1-10), the main obstacle to recovering an accurate $C_{HT}$ is topographic noise (Fig. 10A-H). As we have applied here, and has been previously demonstrated (Hurst et al., 2012; Roering et al., 2007), careful selection of a $\lambda$ sufficiently large to remove such noise, but not so large such that concave valley bottoms introduce positive curvatures, still allows for accurate calculation of $C_{HT}$, particularly for $E^*$=1. By contrast, in cases where $E^*$ is sufficiently high to develop planar side slopes, once $\lambda$ reaches a sufficiently high value to remove topographic noise, the

CWT and PFT kernels have become sufficiently large to incorporate planar slopes into the curvature measurements, thus underpredicting the actual value of $C_{HT}$. In these cases, topographic noise is a secondary impediment to accurate $C_{HT}$ measurement, preventing utilization of a sufficiently small $\lambda$ that avoids planar hillslopes. Furthermore, if $E^*$ is sufficiently large, planar side slopes may appear close enough to the hilltop to disqualify almost any smoothing scale, which is clear from our synthetic hillslopes with no added noise (Fig. 10E, I, M). Importantly, even for $E^*$=10, planar slopes begin to bias $C_{HT}$

(Fig. 10E).

        The grid resolution of digital topographic data has been recognized to affect measurements of topographic curvature and hillslope sediment flux (e.g. Ganti et al., 2012; Grieve et al., 2016b). However, the deviation between known and measured $C_{HT}$ we note here is intrinsic to the form of hillslopes that are described by the nonlinear diffusion model. As $E^*$ increases and the hilltop undergoes a concomitant increase in $C_{HT}$, a smaller $\lambda$ is ideally required to avoid the planer side slopes and accurately

calculate $C_{HT}$. Unfortunately, however, $\lambda$ can only be decreased so much before topographic noise and stochastic and disturbance-driven processes begin to overwhelm the calculated curvature values (Hurst et al., 2012, 2013; Roering et al., 2007; Fig. 3, 10). As such, increasing the resolution of topographic data, while desirable for characterizing hillslope sediment transport processes, will not by itself alleviate the systematic deviation between measured and model-specified $C_{HT}$, as such high-resolution data will also be recording the stochastic signals that deviate from the underlying hillslope form (Roth et al.,

2020). However, improved characterization of the distribution of roughness and microtopography in landscapes and how they may vary with erosion rate may provide a remedy for estimating erosion rate from topography and defining a better-informed $\lambda$, particularly in landscapes where hilltops are conspicuously sharp and where topographic resolution continues to improve.

        Importantly, we stress that neither CWTs nor PFTs are, at this time, capable of accurately estimating hilltop curvature at moderate to high $E^*$, even when $\lambda$ is small (Fig. 10). We observe that the CWT and PFT systematically underpredict $C_{HT}$

when $E^*$=100 (Fig. 10M-P). However, we acknowledge that $E^*$ values of 100 are perhaps unreasonably high for most natural landscapes, with perhaps a few notable exceptions (e.g. Taiwan, Himalaya, New Zealand). More so, soil production limits (e.g. DiBiase et al., 2012; Heimsath et al., 1997; Montgomery, 2007; Neely et al., 2019) imply that these settings may exhibit processes that are not well represented with the soil creep model employed here. Regardless, the CWT and PFT clearly underpredict $C_{HT}$ when $E^*$=30 and exhibit underpredicted $C_{HT}$ when $E^*$=10, *even in the most ideal case when synthetic*

*hillslopes have no added noise*. Similar values of $E^*$ have been recorded in numerous natural landscapes (Clubb et al., 2020; Grieve et al., 2016a; Hurst et al., 2019, 2013, 2012; Marshall and Roering, 2014; Roering et al., 2007). Finally, the addition of topographic noise to our hillslopes serves to increase uncertainty in measurement of $C_{HT}$ (Fig. 10, S4, S7, S10) but does not result in systematic over- or underestimation.

## 5.4 Does hilltop curvature vary linearly with erosion rate?

The systematic underestimation of $C_{HT}$ that we observe here has important implications for interpreting erosion rates and hillslope surface processes in natural soil-mantled landscapes that are not eroding slowly. Specifically, our results here urge caution when applying hilltop curvature measurement techniques to natural soil mantled landscapes eroding at moderate to rapid rates and where hilltops are correspondingly sharp. While $C_{HT}$ has been found to generally agree with independently calculated erosion rates following Equation 1, the measurement artifact we have observed here calls into question the accuracy of calculated erosion rates from $C_{HT}$ in natural landscapes in past studies. Recent observations put forward by Gabet et al. (2021), show that hilltop curvature varies with the square root of erosion rate, which implies a square root relationship between hillslope diffusivity, D, and erosion rate, representing a deviation from the long-held view that $C_{HT}$ varies linearly with erosion rate (Equation 1). Here we use a synthetic hillslope simulation to explore whether these findings may be influenced by a systematic bias in the estimation of $C_{HT}$ due to the formation of planar side slopes proximal to hilltops as described above (Fig. 10). Specifically, we followed the methodology laid out by Gabet et al. (2021) and reproduced a $C_{HT}$-E relationship from synthetic hillslopes with *no added noise*. While Gabet et al. (2021) constructed synthetic hillslope profiles to account for the effect of grid spacing on calculated $C_{HT}$, we additionally consider the role of smoothing scale, λ, on estimation of $C_{HT}$. In order to facilitate comparison, we initially selected λ=14 m, the same scale that Gabet et al. (2021) applied at each of their field sites, to calculate $C_{HT}$. We constructed a series of synthetic hillslopes described by Equation 11 for E*=1-100, which corresponds to erosion rates of ~0.01-1 mm yr$^{-1}$ (for D=0.003 m$^2$ yr$^{-1}$). To constrain how hillslope diffusivity modulates the erosion rate at which $C_{HT}$ is underestimated, we also tested a range of diffusivities, spanning D=0.001-0.005 m$^2$ yr$^{-1}$ (assuming $\frac{\rho_s}{\rho_r}$=0.5). We used the CWT to calculate curvature; a PFT could be used as well, which would be consistent with the Gabet et al. (2021) methodology. However, as we have demonstrated, $C_{HT-P}$ and $C_{HT-W}$ are similar for both natural hillslopes and synthetic hillslopes with no added noise (Fig. 3, 10).

We observe that erosion rate and $C_{HT}$ do not vary linearly as expected from Equation 1 for all erosion rates (Fig. 11). While the relationship between erosion rate and measured hilltop curvature (we plot the absolute value, $|C_{HT}|$, to allow visualization of positive values) is linear as expected from Equation 1 for slow erosion rates of 0.01-0.08 mm yr$^{-1}$ (for the case of D=0.003 m$^2$ yr$^{-1}$), the measured and actual synthetic values of $|C_{HT}|$ begin to clearly diverge for erosion rates >0.08 mm yr$^{-1}$ (Fig. 11A), though some deviation exists at erosion rates as low as ~0.03 mm yr$^{-1}$ (Fig. 11B; blue squares). This deviation occurs at even slower erosion rates for D=0.001 m$^2$ yr$^{-1}$, specifically at erosion rates of ~0.02-0.03 mm yr$^{-1}$ (Fig. 11A). As this deviation increases with erosion rate (and E*), it approximates a square root relationship between erosion rate and hilltop curvature. Importantly, the erosion rate at which this deviation occurs is heavily dependent on smoothing scale and diffusivity. For the range of tested diffusivities (D=0.001-0.005 m$^2$ yr$^{-1}$) and for λ=14 m and λ=20 m, we plotted the ratio of measured $C_{HT}$ to the actual $C_{HT}$ (Fig. 11B). We find that for smaller D, the deviation between measured and model-defined $C_{HT}$ occurs at slower erosion rates, while λ dictates the magnitude of deviation (Fig. 11B). For instance, for D=0.001 m$^2$ yr$^{-1}$ and λ=14 m, we

find that $C_{HT}$ is underestimated by >10 percent for erosion rates >~0.04 mm yr$^{-1}$, and when λ=20 m, $C_{HT}$ is underestimated by >15 percent for that same erosion rate of 0.04 mm yr$^{-1}$. Thus, while the erosion rates at which we observe significant deviation between measured and model-defined $C_{HT}$ tend to be higher than those found in many soil-mantled landscapes (for D=0.003 m$^2$ yr$^{-1}$ and λ=14 m) (Montgomery, 2007), including those tested by Gabet et al. (2021), the strong dependency of this deviation on diffusivity and smoothing scale warrants caution in interpretations of nonlinear relationships between hilltop curvature and erosion rate. Furthermore, without *a priori* knowledge of the diffusivity, determination of the magnitude of $C_{HT}$ underestimation is challenging to ascertain from topographic data alone (Fig. 11B). Importantly, the underestimation in $C_{HT}$ we note here is independent of topographic noise and surface roughness. Incorporation of such roughness will introduce additional uncertainties. We encourage future work to investigate climatic and other factors that dictate hillslope diffusivity and the potential coupling between diffusivity and erosion rate (e.g. Richardson et al., 2019), although care must be taken to ensure that observed relationships do not result from measurement artifacts that deviate from the true underlying hillslope form.

Current hilltop curvature measurement techniques do not have a well-defined capability to filter topographic noise that is inherent to all landscapes and topographic datasets while maintaining an unbiased value of $C_{HT}$ at high values of E* where nearly planar (i.e. negligible curvature) side slopes become increasingly proximal to hilltops. Essentially, the zone of hilltop convexity becomes exceedingly narrow (and thus difficult to estimate) as E* increases. As such, estimates of erosion rates using Equation 1 should be considered minimum erosion rates, particularly in landscapes with conspicuously sharp hilltops (Fig. 10). These results strongly motivate future investigation of the structure of topographic noise in landscapes due to underlying processes such as trees throw and other sources of bioturbation, as well as noise inherent to digital topographic data. Improved understanding of the structure of topographic surface roughness may facilitate future accurate morphologic estimates of erosion rate in moderate to fast-eroding landscapes.

## 6. Conclusions

We utilized 2D continuous wavelet transforms to calculate hilltop curvature in three catchments in the Oregon Coast Range that exhibit a diversity of hillslopes. We found that the measured hilltop curvature values are comparable to those calculated from fitting 2D polynomial functions to topography to calculate curvature, a method that has been commonly applied elsewhere. Both techniques produce estimates of erosion rate that are consistent with those independently constrained from cosmogenic radionuclides in stream sediments. Specifically, we find that erosion rate calculated with the CWT is ~0.156±0.055 mm yr$^{-1}$ in Hadsall Creek, 0.1±0.05 mm yr$^{-1}$ in the North Fork Smith River, and 0.01±0.008 mm yr$^{-1}$ in three small catchments that drain to Bear Creek. We further we find that the 2D continuous wavelet transform operates $10^2$ to >$10^3$ times faster than the 2D polynomial when applied at smoothing scales that are commonly used for calculating hilltop curvature (~5-30 m). We additionally find that the computational advantage of the 2D continuous wavelet transform increases as digital elevation models become larger. This dramatic disparity in operation time opens numerous doors for widespread topographic analysis as high-resolution topographic data becomes increasingly available.

570         We additionally test the accuracy of both the wavelet transform and polynomial by constructing synthetic hillslopes following a nonlinear diffusive hillslope geomorphic transport law. Synthetic hillslopes were constructed with and without added surface noise of various types (white, pink, red/Brownian) and exhibited various forms corresponding to a range of dimensionless erosion rates. We find that both the wavelet transform and polynomial are able to reproduce hilltop curvature for slow dimensionless erosion rates (E*=1-10). However, we also observe that both techniques produce underestimated values

of $C_{HT}$ when E*≥10, as planar hillslopes begin to systematically bias the calculated curvature at the hilltop. While this is in part due to the required smoothing of topography to remove added noise, which in natural landscapes is due to stochastic transport processes as well as noise inherent in digital topographic data, we also find that curvature is underestimated on synthetic hillslopes where there is no added noise. At moderate to high dimensionless erosion rates (E*=30-100), we find that hilltop curvature is systematically underestimated as hillslopes become progressively narrower near the hilltop. This systematic

deviation from the defined and measured hilltop curvature has key implications for predicting erosion rates in soil mantled landscapes. In landscapes eroding at moderate to rapid rates, erosion rates calculated with hilltop curvature should be considered a minimum. Finally, we demonstrate that underestimation of synthetic hilltop curvature at moderate to fast erosion rates results in apparent power law and square root relationships between erosion rate and hilltop curvature. This previously observed relationship from natural hillslopes has led to suggestions that hillslope diffusivity may also vary as the square root

of erosion rate. Our results here, however, demonstrate that this may be a measurement artifact introduced by planar hillslopes biasing hilltop curvature measurements as hilltops progressively narrow and steepen, not simply due to poor data resolution. Future hillslope geomorphic work must more clearly characterize the roughness of soil mantled hillslopes and develop methods that smooth and remove topographic noise while maintaining an unbiased hilltop curvature measurement, if hilltop curvature is to be applied in rapidly eroding landscapes.


**Code and Data Availability**

We utilized TopoToolbox (https://topotoolbox.wordpress.com/download; Schwanghart and Scherler, 2014) code in this paper, which is freely available. We additionally used wavelet codes from the Automated Landslide Mapping toolkit (ALMtools) by

Adam Booth (http://web.pdx.edu/~boothad/tools.html; Booth et al., 2009). Additional MATLAB scripts, including for synthetic hillslope construction, are available at https://github.com/wtstruble. All utilized lidar DEMs are publicly available from the Oregon Department of Geology and Mineral Industries (https://www.oregongeology.org/lidar/).

**Author Contributions**

WS and JR conceived of and designed the study. WS developed the study and completed the analysis. WS prepared the manuscript with contributions from JR.

**Competing Interests**

The authors declare that they have no conflict of interest.


**Acknowledgements**

Thank you to Brooke Hunter, Danica Roth, Fiona Clubb, and Odin Marc for helpful discussions. Tyler Doane, an

anonymous reviewer, and associate editor Simon Mudd provided insightful reviews that improved the quality of the

manuscript. We are particularly grateful to Adam Booth, who provided several helpful and enlightening conversations about

wavelets. [10]Be data processed at Lawrence Livermore National Laboratory.

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

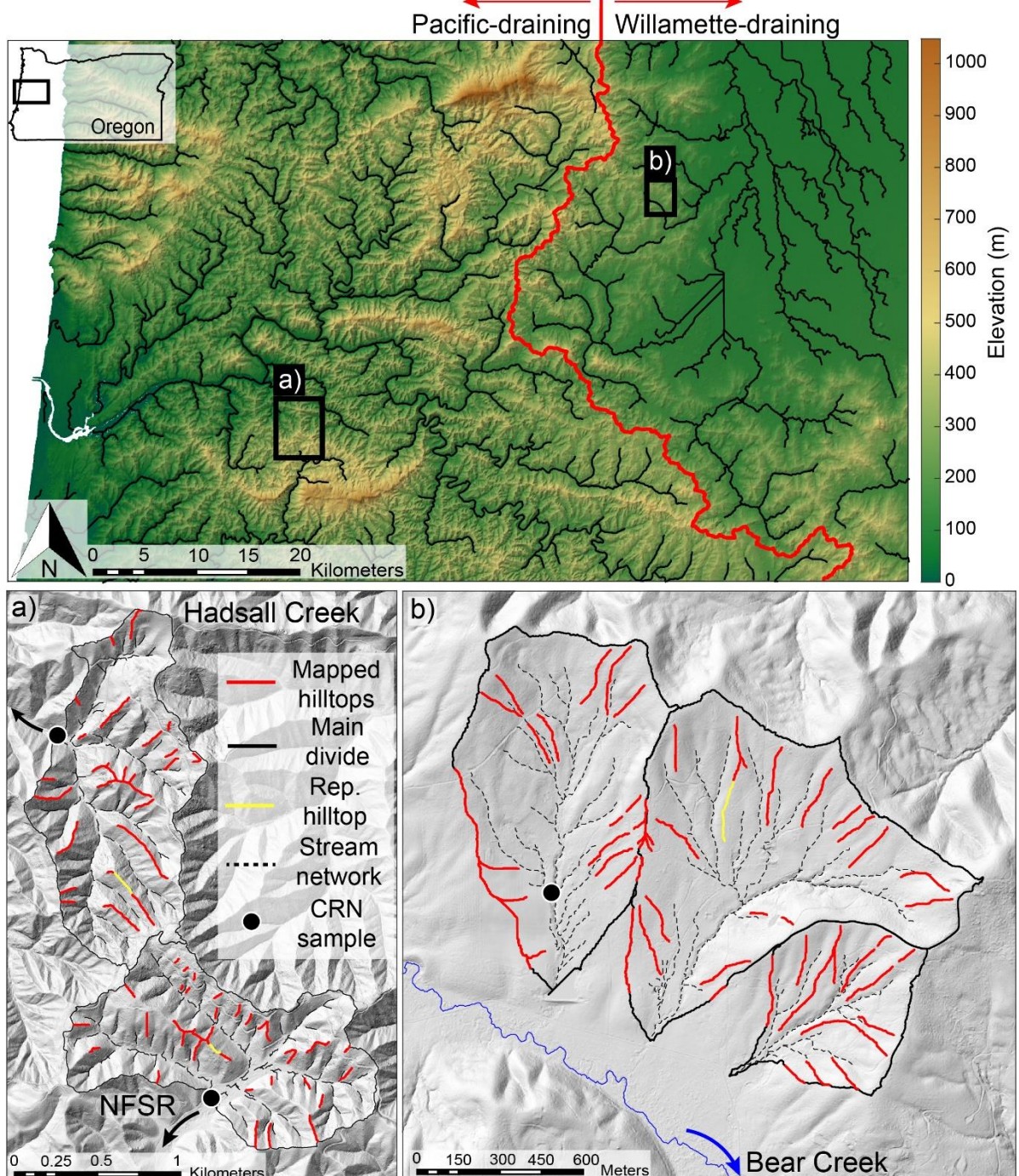

**Figure 1: Oregon Coast Range study sites. Note the drainage divide (red) between catchments that flow directly to the Pacific Ocean and those that flow east into the Willamette River, which then flows northward to the Columbia River. A) Hadsall Creek and the North Fork Smith River (NFSR). B) The three catchments that flow to Bear Creek. Arrows in A) and B) denote river flow direction. Representative (Rep.) hilltop selected so that it approximates the average curvature for the catchment when compared to curvature measurements taken for all hilltops**

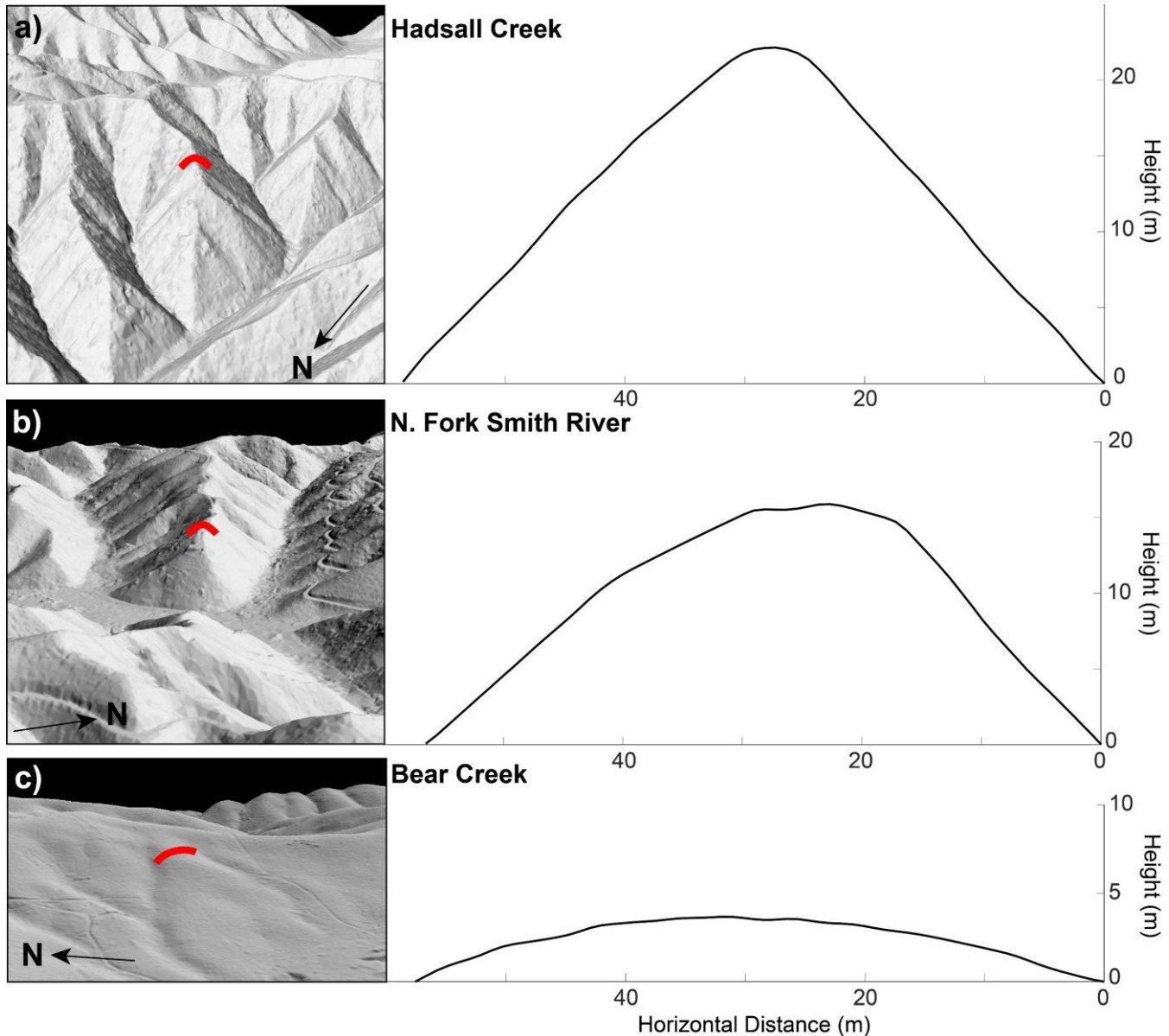

**Figure 2: Oregon Coast Range hillslope profiles.** Example lidar hillshades of hillslopes from Hadsall Creek (a), the North Fork Smith River (b; NFSR), and Bear Creek (c). Red lines in hillshades correspond to the hillslope profiles in right column. Note that hillslope profiles have the same horizontal scale, allowing for clear visualization of the difference in hillslope relief between sites. Each sample hillslope profile corresponds to the representative hilltop in each catchment (yellow lines in Figure 1). Note that at the rapidly eroding Hadsall Creek and NFSR, the hillslopes have attained threshold gradients and are near-planar. $C_{HT}$, however, still reflects the difference in erosion rate between sites.

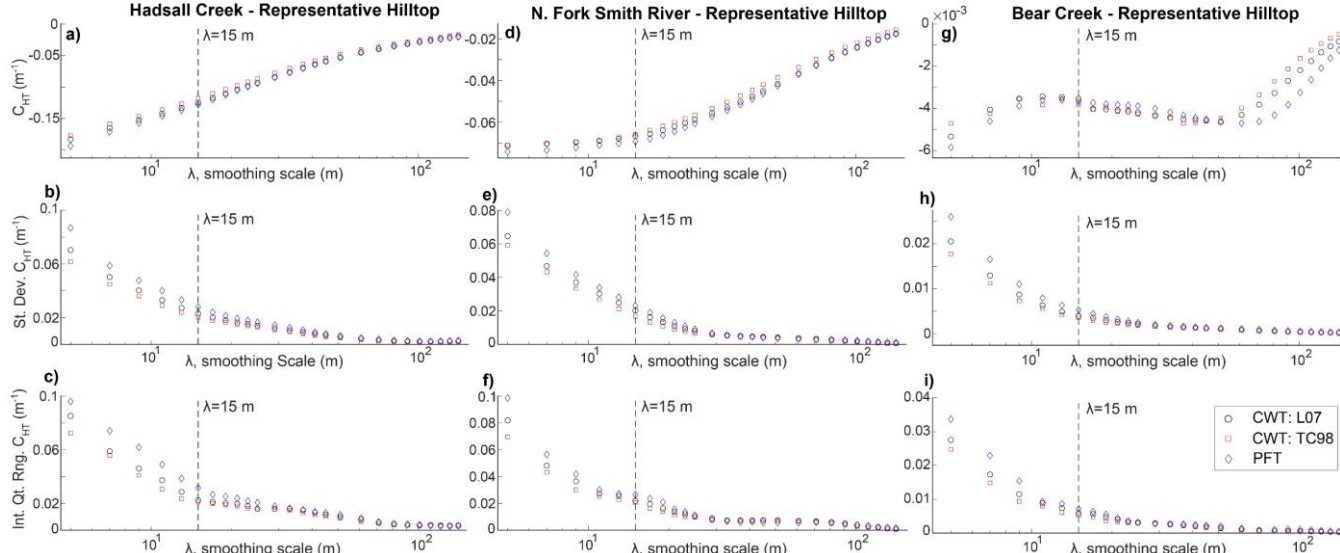

Figure 3: Curvature extracted from representative hilltop at Hadsall Creek, NFSR, and Bear Creek for a range of λ. Upper row is $C_{HT}$ measurements, second row is the standard deviation of $C_{HT}$, and the bottom row is the interquartile range of $C_{HT}$. Note that the scaling break that identifies where tree throw pits are filtered out depends on the size of the considered hillslope and consistency of pit-mound topographic in a landscape. Here, though, a break exists at λ≈15 m for Hadsall Creek (especially apparent in standard deviation and interquartile range) and the NFSR and Bear Creek at λ≈11-20 m (note that second break at ~60 m in Bear Creek corresponds to the introduction of concave valleys). These scaling breaks are generally consistent with those observed for the OCR by Roering et al. (2010) and are visible for CWT and PFT λ definitions. L07: Lashermes et al. (2007); TC98: Torrence and Compo (1998).

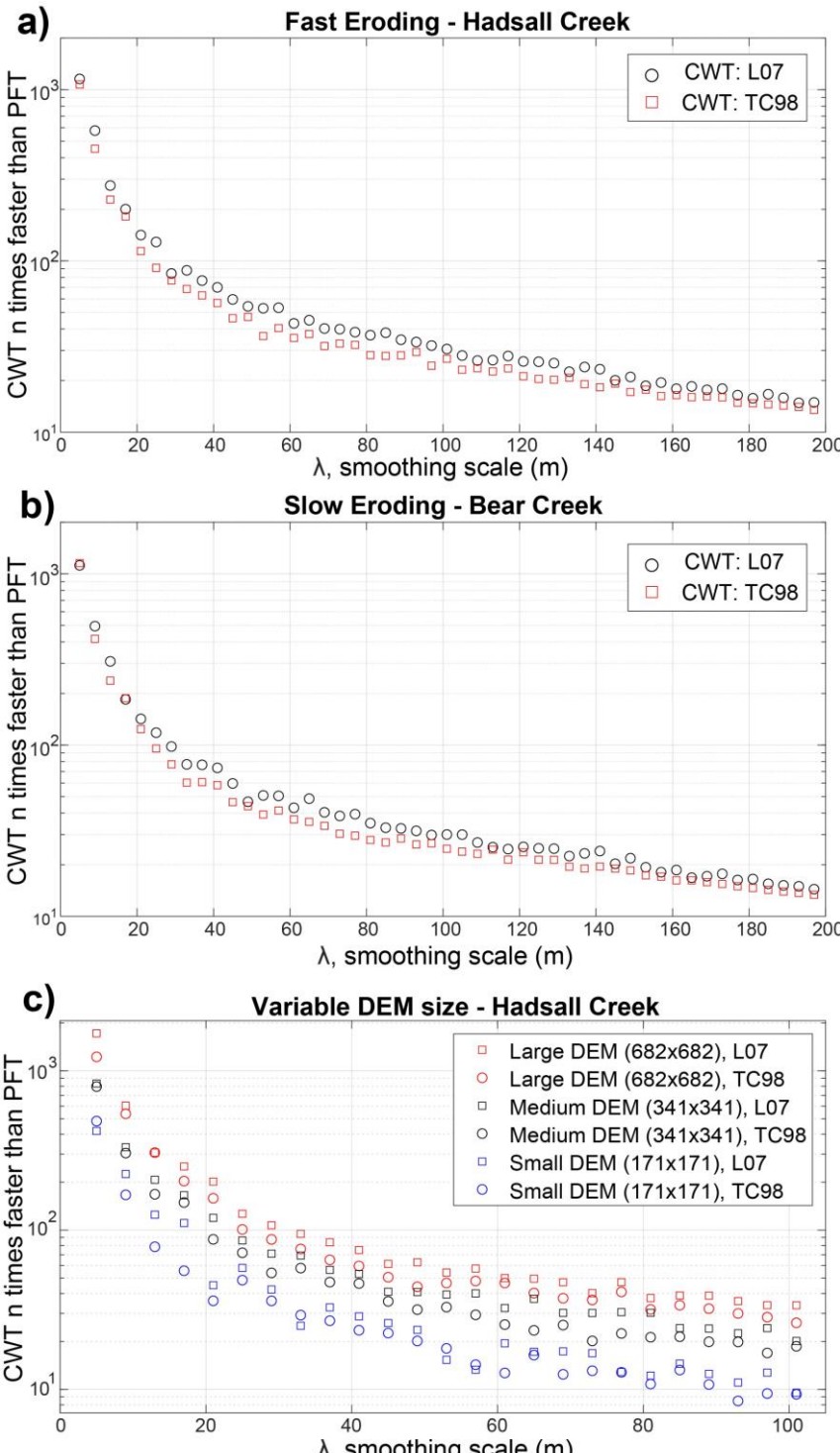

**Figure 4: Speed of the CWT compared to the PFT for different smoothing scales, λ. A, B: Relative speed of the CWT to the PFT for small portions (513x513 single precision grid, cell size of 0.9144 m) of the Hadsall and Bear Creek**

catchments, quantified as the ratio of CWT/PFT processing time. Thus, for each smoothing scale, each point can be interpreted as the CWT being *n* times faster than the PFT. At small λ, the CWT is >1000 times faster than the PFT. The CWT remains >100 times faster than the PFT until λ≈30 m, a scale that is usually larger than most smoothing scales utilized in C$_{HT}$ calculation. C: Relative speed of the CWT to PFT for DEMs of various size in Hadsall Creek for λ=5-101 m. Largest DEM is 682x682 pixels. The medium-sized DEM is the upper-left quadrant of the large DEM (341x341 pixels), and the small DEM is upper-left quadrant of medium DEM (171x171 pixels). Note that the CWT increases in relative speed as DEM size increases. L07: Lashermes et al. (2007); TC98: Torrence and Compo (1998).

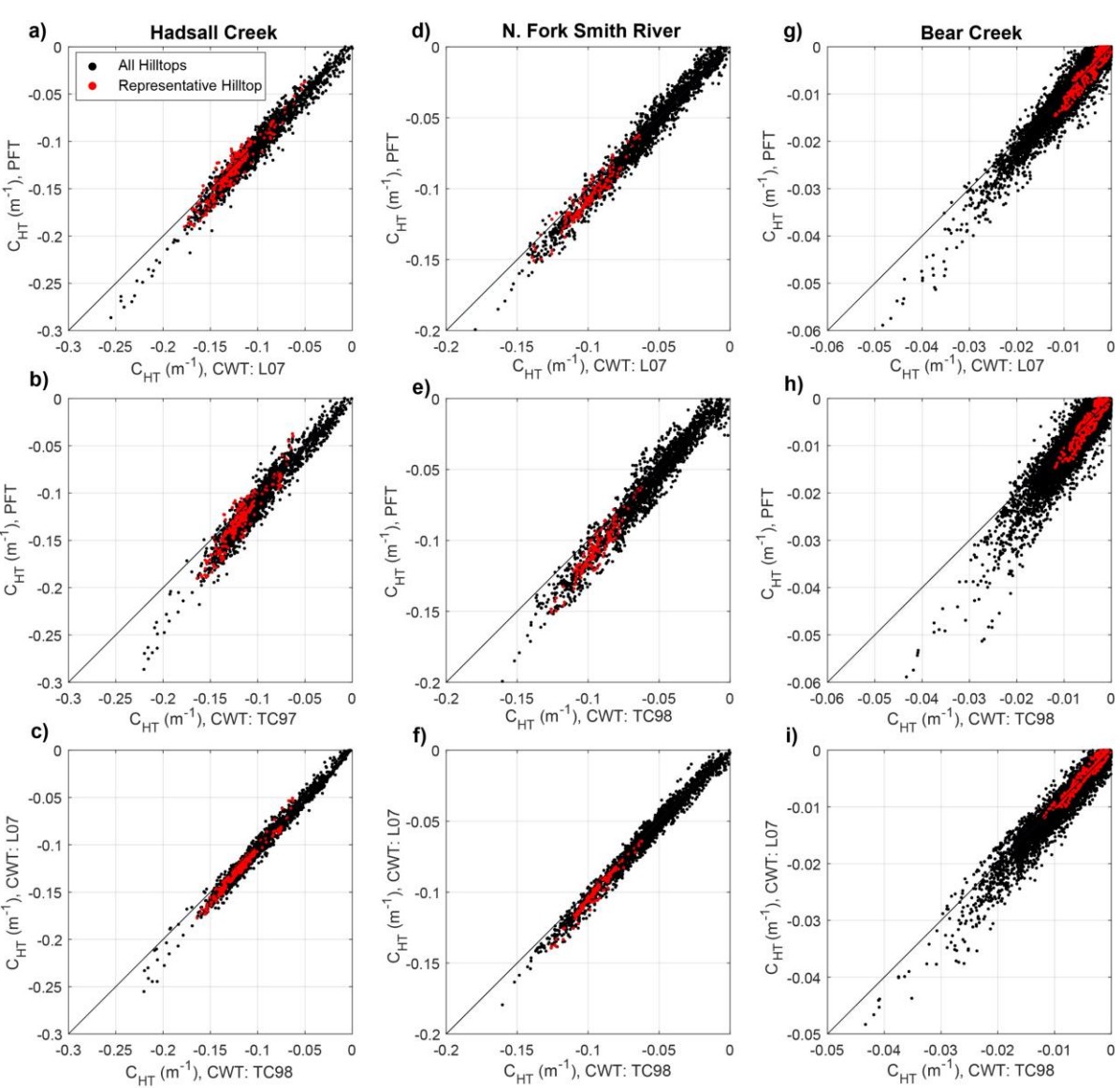

**Figure 5: Comparison of C$_{HT}$ calculation methods for λ=15 m. Black dots correspond with curvature measured at nodes for all mapped hilltops (roads, landslides, valley bottoms, etc. *removed*). Red points correspond with the representative**

hilltop nodes (Fig. 1). Perfect agreement between measurement techniques would plot as 1:1 line (black line). Recall that more positive (lower magnitude) $C_{HT}$ corresponds with more gentle hillslopes (upper-right corner). See text for details. L07: Lashermes et al. (2007); TC98: Torrence and Compo (1998).

| Site | Lat. (° N) | Long. (° W) | | Mean $C_{HT}$ (m$^{-1}$) | | | Median $C_{HT}$ (m$^{-1}$) | | | Standard Deviation $C_{HT}$ (m$^{-1}$) | | |
|---|---|---|---|---|---|---|---|---|---|---|---|---|
| | | | | CWT (L07) | CWT (TC98) | PFT | CWT (L07) | CWT (TC98) | PFT | CWT (L07) | CWT (TC98) | PFT |
| Hadsall Creek | 43.983 | -123.823 | All Hilltops | -0.104 | -0.099 | -0.110 | -0.111 | -0.106 | -0.116 | 0.037 | 0.034 | 0.040 |
| | | | Rep. Hilltop | -0.125 | -0.119 | -0.129 | -0.126 | -0.120 | -0.128 | 0.023 | 0.020 | 0.028 |
| NFSR | 43.963 | -123.811 | All Hilltops | -0.061 | -0.059 | -0.065 | -0.059 | -0.057 | -0.061 | 0.033 | 0.037 | 0.037 |
| | | | Rep. Hilltop | -0.067 | -0.066 | -0.069 | -0.063 | -0.063 | -0.068 | 0.020 | 0.017 | 0.023 |
| Bear Creek | 44.181 | -123.371 | All Hilltops | -0.007 | -0.006 | -0.008 | -0.006 | -0.005 | -0.006 | 0.006 | 0.005 | 0.007 |
| | | | Rep. Hilltop | -0.005 | -0.005 | -0.006 | -0.004 | -0.004 | -0.005 | 0.003 | 0.003 | 0.004 |

**Table 1: $C_{HT}$ measured at OCR study sites by the CWT and PFT for λ=15 m. L07: λ definition of Lashermes et al. (2007); TC98: λ definition of Torrence and Compo (1998). All values rounded to nearest 10-thousandth.**


| Site | Lat. (° N) | Long. (° W) | | Mean E (mm yr$^{-1}$) | | | Median E (mm yr$^{-1}$) | | | Standard Deviation E (mm yr$^{-1}$) | | | CRN E (mm yr$^{-1}$) |
|---|---|---|---|---|---|---|---|---|---|---|---|---|---|
| | | | | CWT (L07) | CWT (TC98) | PFT | CWT (L07) | CWT (TC98) | PFT | CWT (L07) | CWT (TC98) | PFT | |
| Hadsall Creek | 43.983 | -123.823 | All Hilltops | 0.156 | 0.149 | 0.164 | 0.167 | 0.159 | 0.174 | 0.055 | 0.051 | 0.060 | 0.113±0.018[a] |
| | | | Rep. Hilltop | 0.188 | 0.178 | 0.193 | 0.189 | 0.179 | 0.193 | 0.034 | 0.030 | 0.042 | |
| NFSR | 43.963 | -123.811 | All Hilltops | 0.100 | 0.099 | 0.104 | 0.088 | 0.086 | 0.092 | 0.050 | 0.046 | 0.055 | 0.058±0.0054[a] |
| | | | Rep. Hilltop | 0.092 | 0.088 | 0.097 | 0.095 | 0.095 | 0.101 | 0.030 | 0.025 | 0.035 | |
| Bear Creek | 44.181 | -123.371 | All Hilltops | 0.010 | 0.009 | 0.012 | 0.008 | 0.008 | 0.009 | 0.008 | 0.007 | 0.010 | 0.008±0.0007[b] |
| | | | Rep. Hilltop | 0.007 | 0.007 | 0.009 | 0.006 | 0.006 | 0.008 | 0.005 | 0.005 | 0.006 | |

**Table 2: Erosion rate at OCR study sites calculated with Equation 1, assuming D=0.003 m$^2$ yr$^{-1}$ and $\frac{\rho_s}{\rho_r}$=0.5. L07: λ definition of Lashermes et al. (2007); TC98: λ definition of Torrence and Compo (1998). All values rounded to nearest 10-thousandth.**
**[a]Recalculated erosion rates from Penserini et al. (2017); see Table 3.**
**[b]See Table 4.**


| Catchment | Location | Concentration (atoms g$^{-1}$ quartz) | Error (atoms g$^{-1}$ quartz) | Erosion Rate (mm/yr) | Error (mm/yr) | Notes |
|---|---|---|---|---|---|---|
| Hadsall Creek | 43.985°N, -123.824°W | 33766.10 ($^{10}$Be) | 4666.26 ($^{10}$Be) | 0.113 | 0.018 | Recalculated from Penserini et al. (2017) |
| NFSR | 43.964°N, -123.811°W | 70902.91 ($^{10}$Be) | 3408.59 ($^{10}$Be) | 0.058 | 0.0054 | Recalculated from Penserini et al. (2017) |

**Table 3: Recalculated CRN erosion rates**
**We used the CRONUS online calculator (Balco et al., 2008) to determine catchment-averaged erosion rates from $^{10}$Be in stream sediment. The samples from Hadsall Creek and NFSR are recalculated from the $^{10}$Be data presented by Penserini et al. (2017). Reported CRN error is from external uncertainty.**


| Catchment | Location | Mean Elevation (m) | Shielding Factor | Quartz Weight[a] (g) | Be Carrier Weight (mg) | $^{10}$Be/$^9$Be (x 10$^{-13}$) | $^{10}$Be Concentration (atoms g$^{-1}$ quartz) | Erosion Rate (mm/yr) |
|---|---|---|---|---|---|---|---|---|
| Bear Creek | 44.186°N, -123.375°W | 240 | 1 | 25.02 | 221 | 6.338±0.1175 | 400833±8011.37 | 0.008±0.0007 |

**Table 4: New CRN erosion rate at Bear Creek**
**[a]Assumed a density of 2.6 g/cm$^3$.**
**We used the CRONUS online calculator (Balco et al., 2008) to determine catchment-averaged erosion rates from $^{10}$Be in stream**
**sediment at Bear Creek. Reported CRN error is from external uncertainty.**

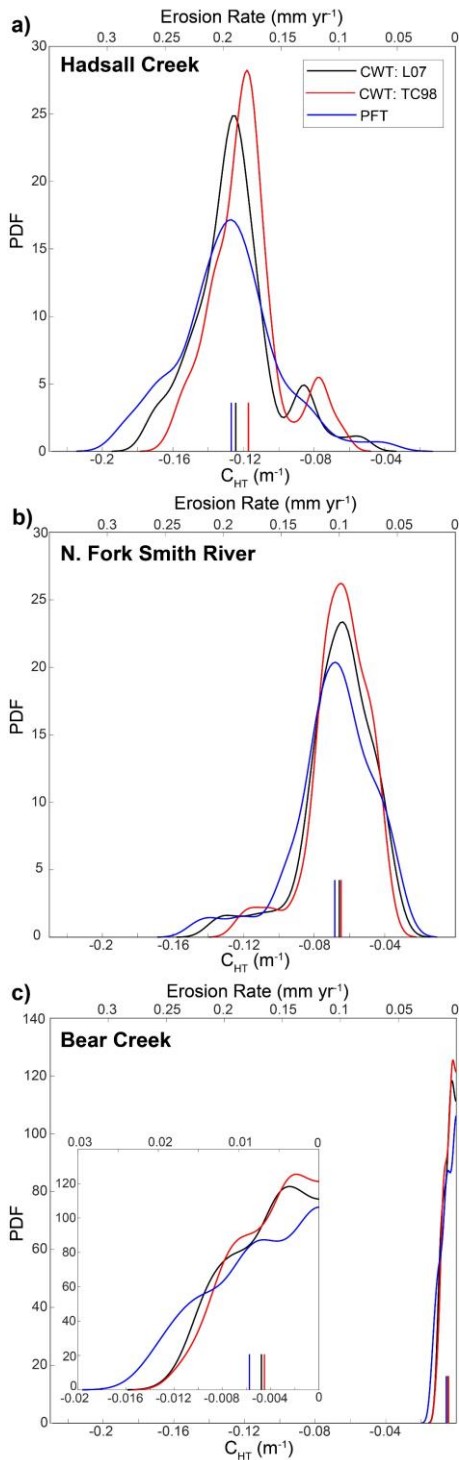

**Figure 6: Probability density functions of $C_{HT}$ (bottom x-axis) and erosion rate calculated using Equation 1 (top x-axis) for the representative hilltop at each OCR field site. See Fig. S1 for all mapped hilltops version. Note agreement between each $C_{HT}$ calculation method. Further, note dramatic variability in $C_{HT}$ between sites (all panels use same x-axis; inset**

in panel C more clearly displays distribution of $C_{HT}$ at Bear Creek). Small vertical lines at bottom of each panel represent the mean of the plotted distribution (Table 2). Note that positive $C_{HT}$ values are not permitted in the output PDF (C). L07: Lashermes et al. (2007); TC98: Torrence and Compo (1998).









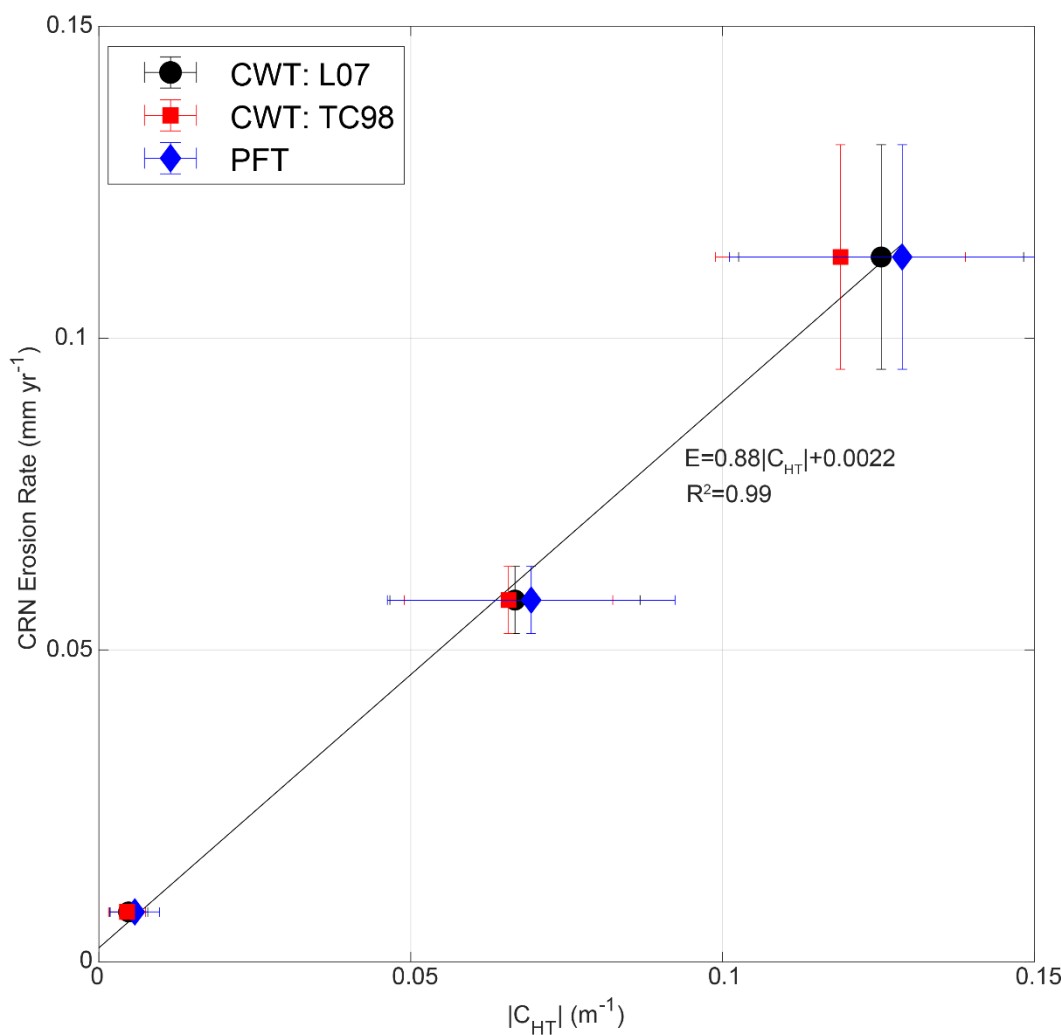


**Figure 7: CRN erosion rate vs $C_{HT}$. CRN erosion rates for Bear Creek (slow E), NFSR (moderate E), and Hadsall Creek (fast E) against the absolute value of $C_{HT}$ for the representative hilltop in each catchment. Filled symbols are mean E and $C_{HT}$ values and errorbars correspond to the standard deviation of $C_{HT}$ and external uncertainty in CRN erosion rate measurements (Table 1, 3). Note that errorbars may be smaller than the size of the mean symbol for Bear**
**Creek samples. L07: Lashermes et al. (2007); TC98: Torrence and Compo (1998).**

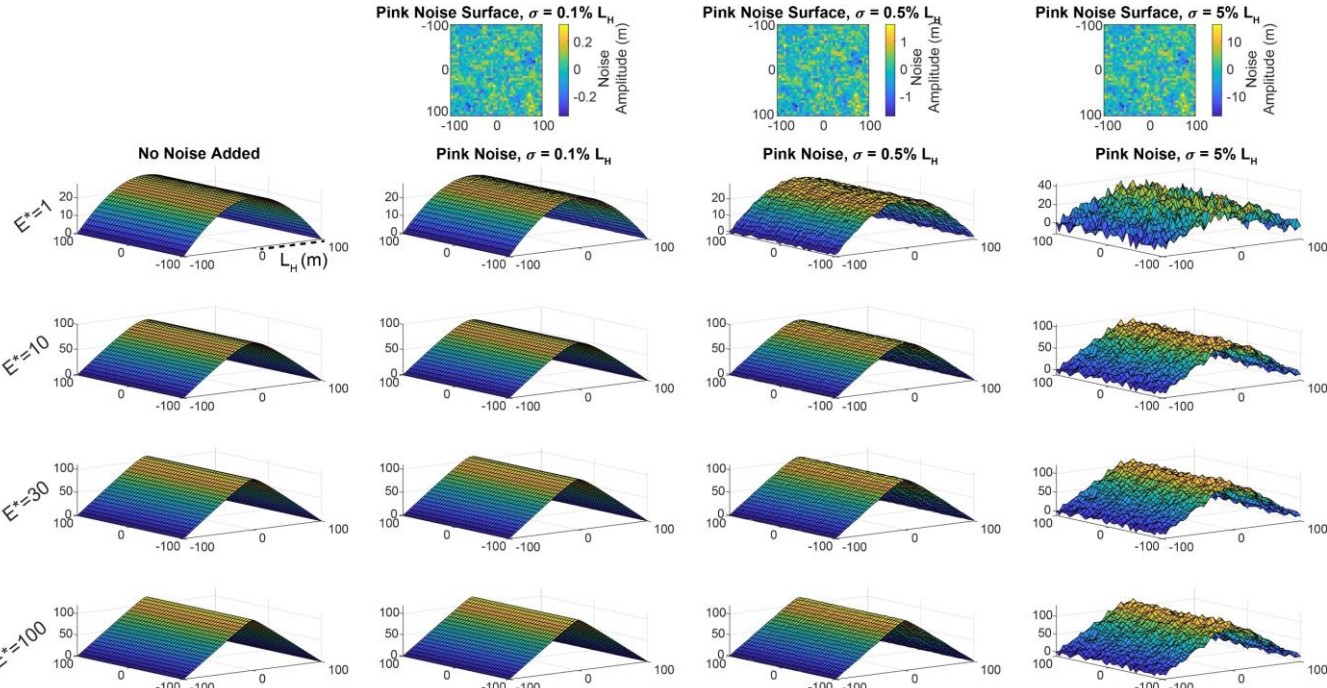

Figure 8: Synthetic hillslopes constructed using Equation 11. Upper row shows pink noise surfaces that are added to the original hillslope form (left column); yellow colors correspond with positive deviations from the hillslope (convex noise) and blue with negative deviations (concave noise). Each row of hillslopes corresponds with range of dimensionless erosion rates, from E*=1-100. Note the increased prominence of planar hillslopes as E* increases; the z-axis on each plot may differ. Noise does not vary with E*; thus the magnitude of noise relative to hillslope relief is more visually apparent at lower E* (See σ=5% $L_H$ column for clear example). Note that all results in Fig. 10C, G, K, and O correspond with the third column here (σ=0.5% $L_H$). See supplemental for corresponding figures for σ=0.1%$L_H$ and σ=5%$L_H$ cases.

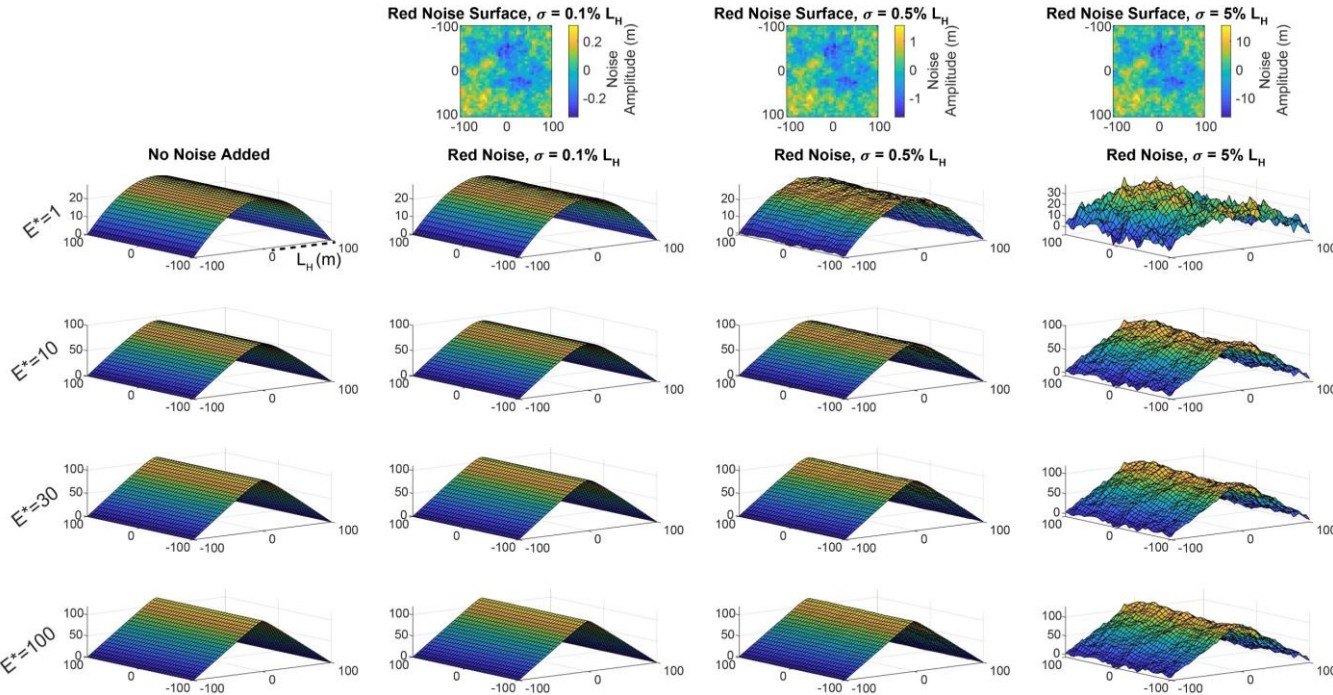

**Figure 9: Synthetic hillslopes constructed using Equation 11. Same as Fig. 8, but with red noise added (see supplemental for white noise example). Upper row shows red noise surfaces added to the original hillslope form (left column); yellow colors correspond with positive deviations from the hillslope (convex noise) and blue with negative deviations (concave noise). Each row of hillslopes corresponds with dimensionless erosion rates from E\*=1-100. Note the increased prominence of planar hillslopes as E\* increases. Noise does not vary with E\*; thus the magnitude of noise relative to hillslope relief is more visually apparent at lower E\* (See σ=5% L$_H$ column for clear example). Note that compared to Fig. 8, the surface noise exhibits longer wavelength noise, made apparent by larger concave and convex regions. Note that all results in Fig. 10D, H, I, and P correspond with the third column here (σ=0.5% L$_H$). See supplemental for corresponding figures for other σ=0.1%L$_H$ and σ=5%L$_H$ cases.**

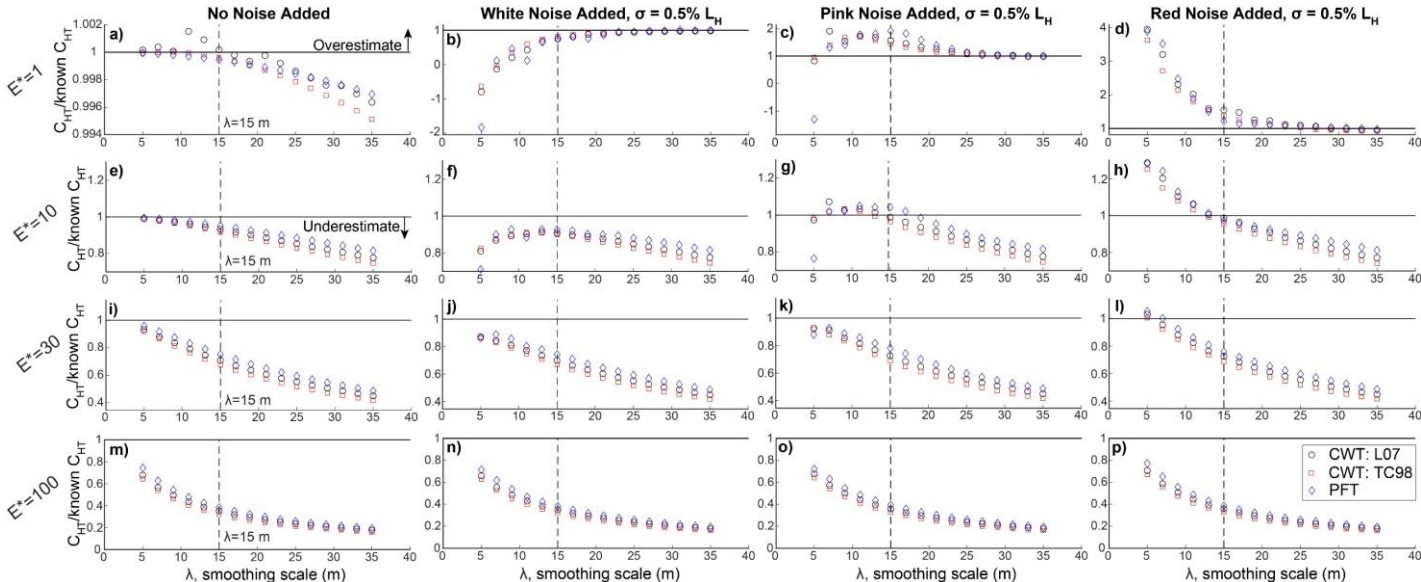

**Figure 10: Ratio of $C_{HT}$ of synthetic hillslopes where E\*=1, 10, 30, and 100 measured at various smoothing scales, λ, with: no noise added (first column), σ=0.5% $L_H$ white noise (second column), pink noise (third column), and red (Brownian) noise (fourth column). Ratio of $C_{HT}$ is quantified as the quotient of the $C_{HT-W}$ or $C_{HT-P}$ and the model-specified $C_{HT}$. Black horizontal line in each panel corresponds with where the measured $C_{HT}$ equals the actual synthetic $C_{HT}$ (i.e. ratio=1). Points that plot above the line correspond with locations where $C_{HT}$ is overestimated (sharper hilltops than expected); points that plot below are underestimations (broader hilltops that expected). See text for details but note systematic underestimation of $C_{HT}$ as E\* increases, even for the surface with no added noise. Dashed vertical line indicates λ=15 m. Note that y-axis differs between E\*=1 plots. L07: Lashermes et al. (2007); TC98: Torrence and Compo (1998).**

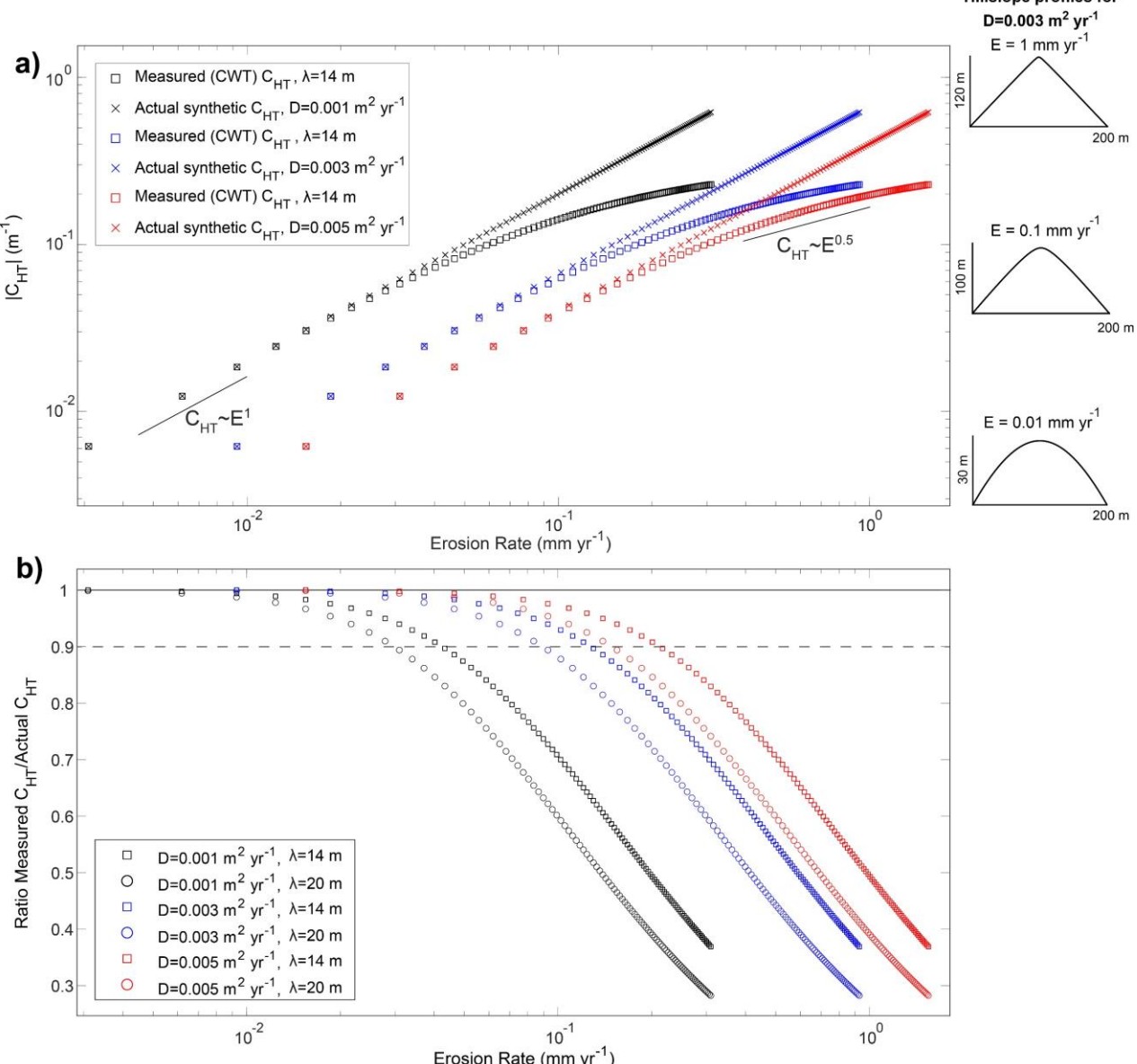

**Figure 11: A) The absolute value of C$_{HT}$ plotted against erosion rate for synthetic hillslopes constructed for E\*=1-100, corresponding to erosion rates of ~0.01-1 mm yr$^{-1}$ for diffusivities ranging from D=0.001-0.005 m$^2$ yr$^{-1}$ ($\frac{\rho_s}{\rho_r}$=0.5). Crosses correspond with the actual C$_{HT}$ for each synthetic hillslope constructed for a given E\*. Squares are C$_{HT-W}$, using the** L07 λ definition. **In this case λ = 14 m. Note the linear relationship between C$_{HT}$ and erosion rate at small erosion rates, in agreement with Equation 1. As erosion rate increases, the relationship between measured C$_{HT}$ and E is no longer linear but could be potentially expressed as a power law. The erosion rate at which this deviation occurs is visualized in panel B. An example square root relationship is plotted at these erosion rates for reference (Gabet et al., 2021). Note example synthetic hillslopes profiles (for D=0.003 m$^2$ y$^{-1}$) spanning the range of erosion rates on right side of panel (y-**

**axes of profiles differ). B) Ratio of measured $C_{HT}$ and the actual model-defined $C_{HT}$ for synthetic hillslopes constructed for $E^*$=1-100 using a range of diffusivities (D=0.001-0.005 m$^2$ yr$^{-1}$, $L_H$=100 m) and measured with $\lambda$=14 m (same as panel A) and $\lambda$=20 m. For each case, the measured $C_{HT}$ deviates from the known value as erosion rate increases. The erosion rate at which this deviation occurs depends on diffusivity and smoothing scale, $\lambda$. Dashed line corresponds to where $C_{HT}$ is underestimated from the true value by 10 percent.**