# Peer review of "Hilltop curvature as a proxy for erosion rate: Wavelets enable rapid computation and reveal systematic underestimation"

_Earth Surface Dynamics, 2021_

## Referee Comment (RC1)

Tyler Doane

Review for **Struble and Roering, Hilltop curvature as a proxy for erosion rate: Wavelets enable rapid computation and reveal systematic underestimation**.

The authors present an application of wavelets to calculate the ridgetop curvature. Quantifying erosion rates is one of the key elements and biggest challenges of geomorphology. Recent work suggests that calculating ridgetop curvature is one of only a handful of topographic measurements that can do so. The increasing availability of high resolution lidar datasets makes this a particularly appealing method for obtaining erosion rates. However, topography is noisy and complicates accurate measurements of ridgetop curvature that reflect that background erosion rates. Wavelets serve as a relatively fast way to simultaneously perform a low-pass filter that removes topographic noise and calculate curvature for that smoothed surface.

The authors present a nice development of ridgetop curvature as a metric for erosion rates, introduction to wavelets, application of the method, and exploration of various scenarios with noisy surfaces. Wavelets have previously been used to calculate curvature across landscapes (Lashermes et al., 2007). However, this is the first application and consideration of the method to ridgetop curvature. The distinction of ridges is key because the smoothing scale of wavelets changes the curvature values - particularly when the ridge is sharp. The authors identify that a smoothing scale of roughly 15 meters is appropriate for the Oregon Coast Range and add to a body of work that demonstrates general agreement between long term erosion rates and hilltop curvature. The authors also explore and suggest an explanation for a recently observed relationship between measurements of ridgetop curvature and erosion rates that deviate from the expected linear relationship.

I believe that this work is quite publishable following a few changes. Most of my comments are relatively minor, but I have a few points (mostly concerning the presentation of the wavelet) that I think might add clarity.

**General Comments**:

**Organization**: I think that the paper might read more smoothly if there's a bit of rearrangement. I suggest that the authors first describe all methods of calculating curvature then move on to field demonstration. It currently jumps from methods, to field setting, and back to methods. I suggest that the authors place 3.1 directly after the introduction and move the description of the field site to directly before the application to watersheds. I also wonder if the authors might combine the two sections on computational efficiency in to one section - probably section 4.1. Section 3.2 is just rather short and straight-forward.

**Torrence and Compo**: To me, the Torrence and Compo wavelet is a distraction throughout the paper. The wavelet that is introduced by Lashermes et al (2007) is explicitly for calculating the second derivative, whereas the wavelet in Torrence and Compo is not. The relationship between $\lambda$ and $s$ in L07 is much more intuitive for this purpose then. Further, I am not convinced the results between the wavelet definitions for a single value of $\lambda$ are comparable.

**DoG wavelets**: I think it would be clearer if the wavelets were presented as the second derivative of a specific Gaussian. The Gaussian can be thought of as the kernel for the low-pass filter,

which I find to be a much more intuitive thing. Further, because it is a low-pass filter, the Gaussian must sum to unity and so the parameter, $s$, has a very specific meaning. I suggest starting with the Gaussian and then presenting the second derivative written out with $s$ in the expression.

I have several line comments throughout - some of which are specific examples of the comments suggested above.
* * *
**line comments**:

**line 33**: Aren't landscapes composed entirely of rivers and hillslopes?

**line 36**: Boundary conditions? It's not clear that these are boundary conditions

**line 51**: Why the quotes around 'nonlinear'? I understand they are getting to a nonlinear flux formulation, which drives a nonlinear response of hillslopes. I suggest rearranging and stating that nonlinear flux relationships with slope drive hillslope responses that become insensitive to further increases in erosion rate.

**line 56**: "recorder" seems unusual to me. "soil mantled hillslopes are an effective record of" would work.

**lines 73-85**: The simplest way to calculate or estimate the curvature is to use the central difference. I recognize that this is sensitive to topographic roughness - but it seems like it should be addressed before moving on to more involved methods. Further, even without smoothing, I wonder how much curvature calculations would differ between the central difference and wavelet methods if one were to perform the central difference on elevations that were, say, 10 meters apart as opposed to the one meter that lidar gets you.

**line 74**: How is curvature defined multiple ways? it is always the second derivative of topography right?

**line 80**: Change to something like: "To reduce to error introduced by topographic roughness..."

**line 80**: "... stochastic sediment perturbations ..." I suspect that the authors meant to say "stochastic sediment transport perturbations". But even so, I think that they should remove the reference to sediment transport because there are other sources of topographic noise - boulders, lithology, hollows, etc.

**line 157**: I think that this statement needs a citation or a mathematical justification. Maybe just state the property of derivatives of convolutions as done in Lashermes et al., 2007?

**line 157**: Based on the rules for derivatives of convolutions it seems like curvature is calculated with the positive second derivative. Why do you use the negative here?

**Eqs. 7 and 8**: I find these definitions a bit confusing. It seems contradictory that $\lambda$ is a physical

thing but has two different definitions with regard to $s$. Since $s$ is the parameter for the wavelet, I imagine setting this equal to a single value and plugging it in to (6), which would produce one result but would have been applied over two different scales according to these definitions? I think that some clarity might come from writing out the function of the original Gaussian that forms the basis for the DoG, because then the meaning of $s$ becomes clear. The original Gaussian must sum to unity because it is the kernel in a low-pass filter and therefore, $s = 2\sigma^2$, where $\sigma$ is the characteristic length scale (or standard deviation) of a Gaussian. This is consistent with Lashermes et al., 2007 and clearly leads to definitions of the second derivative of topography. The definition of wavelets from Torrence and Compo is perhaps more general and is capable of relating the energy contained in certain wavelengths at certain positions in the landscape, but it is not clear that it directly computes curvature. I think that it would be simpler to acknowledge Torrence and Compo and other applications of wavelets or definitions, but just consider the case that is more clearly linked to the second derivative.

**Section 3.3** How is the calculation performed? with Fourier transforms? Isn't that relevant for the computational speed?

**Figure 3**: This seems like two different ways to display the same information. I would suggest using c and d because the raw time - i suspect depends more on the users computational power whereas I would expect the ratio of the run times to remain similar for different users (so long as the domains are the same size).

**Figure 4**: Is this full page figure necessary? I'm not sure that it adds much more to the manuscript. **section 4.1**: How do the run times scale linearly with domain size? The real value in this seems to be when one would apply it over a much larger domain.

**line 290**: See comment of equations 7 and 8. I think that this continues to add unnecessary details and confusion. For a single value $s$, we get two different smoothing scales, which implies that you are not using the same wavelet and I think that the two are not directly comparable. In one, the length-scale of the original Gaussian is multiplied by a factor of $\sqrt{5/2}$ of the other.

**Figure 5C**: The mode of curvature is zero. Which suggests erosion rate of zero on most of the watershed. Is this an issue?

**Figure 6**: Great results!

**line 328**: standard deviation of 50 cm seems quite large. Roth et al., median roughness values approach 10 cm.

**Figure 7**: add "Note the z-axis on each plot may differ".
**line 410**: How does the result change by calculating curvature via wavelets with $\lambda = 15$ m on a 1-m resolution lidar dataset versus, say, a 10 meter resolution DEM which are available for the contiguous U.S? I understand that a wavelet with a smoothing scale of 15 meters is taking a weighted average over 15 meters of topography and calculating the second derivative at 1 meter. But is the result appreciably different from just using a 10 meter DEM to begin with?

**line 427**: This is the first time that the authors have brought up linear diffusion and the steady state profile of topography. I think the message would still come across if they were to remove the reference and reword to say that slow eroding landscapes have broader ridges where the curvature is finite and represents background geomorphic conditions.

**line 530**: power law and square root relationship?

---

## Author Comment (AC1)

Revisions made to Struble and Roering based on comments by Reviewers 1 and 2. All line number references refer to numbering in original manuscript (as referenced by reviewer).

**Reviewer 1, Doane:**

**General Comments:**

**Organization:** We appreciate this point, as we tried several different organizational schemes. We prefer to keep the material in its current order, as it allows us to introduce the polynomial calculation as previous work (it is essentially the primary past work to which we compare the CWT), introduce the field site, and then discuss the new methods (CWT for  $C_{HT}$ ), without getting sidetracked by the field site later.

**Torrence and Compo:** Perhaps we are misunderstanding, because we disagree. Torrence and Compo explicitly define the derivative of a Gaussian, but in a very slightly different functional form (see the  $\frac{d^m}{d\eta^m}$  term and following in their Table 1. The leading term with the gamma function defines the amplitude; *m* refers to the order of the derivative. We use *m*=2.), but the shape of the function is the same. The primary difference is in the definition of wavelet scale. Lashermes et al. define the scale as the inverse of the function's band-pass frequency while Torrence and Compo plug a cosine function with a known frequency into the wavelet basis function (the DoG) to calculate the wavelet transform and identify the scale, *s*, at which the wavelet power spectrum reaches its maximum. Admittedly, we prefer the simplicity and more intuitive definition of Lashermes et al. (and we do not explicitly go into details on *how* Lashermes et al. and Torrence and Compo define  $\lambda$  to avoid introducing too much confusion). Yet, both are valid. So, for the sake of thoroughness, we will maintain usage of Torrence and Compo.

In regards to the question are "the results between the wavelet definitions for a single value of  $\lambda$  comparable?" We don't see why they wouldn't be. We aren't mixing the definitions and then completing a single analysis (and we aren't using the definitions to construct a wavelet with a different functional form, either). For a given  $\lambda$ , we use equations 7 and 8 (now eqns 9 and 10) to calculate wavelet scales, *s*, for both the Lashermes et al. and Torrence and Compo definitions, to then apply the wavelet transform (separately, once for each definition of *s*). It is only after extracting CHT that we compare the two. In essence, in addition to comparing to broad methodologies (CWT to PFT), we are comparing two separate CWT methods as well.

We have opted, then, to not make any substantial adjustments regarding this comment. At first it may seem a bit of a distraction, and we struggled with the best way to present these two wavelet scale definitions. We feel, though, that it is important to acknowledge that both definitions are reasonable as they produce similar results and are both mathematically acceptable as far as we are aware.

**DoG wavelets:** This is a fair point. We have adjusted this section to include an equation for the Gaussian and we have also recast the equation for the Ricker wavelet such that it includes *s*. We've also taken advantage of this adjustment to discuss more carefully the property of convolutions that allows for simultaneous low pass filtering and curvature calculation.

Line comments:

Line 33: Removed phrase.

Line 36: Changed to "landscape properties."

Line 51: Modified these few sentences.

Line 56: Replaced "recorder" with "record."

**Lines 73-85:** We appreciate this comment, but it largely ignores much of the previous work done that demonstrates the importance of using more involved curvature calculation techniques as well as the necessity of using high-resolution lidar data, as opposed to 10-m data, which are not of sufficient quality to measure curvature (even robust statistics cannot necessarily get around the data quality issues with 10-m data compared to lidar). Sources such as Grieve et al. (2016), Passalacqua et al. (2010), and Ganti et al. (2012) are a few examples (we cite them in the paper). Our enhanced discussion of the convolution hopefully helps this discussion some.

Line 74: Adjusted to "types of curvature (i.e. tangential, planform, Laplacian, etc.)"

Line 80: We've adjusted to read "reduce the impact of topographic roughness," as it isn't always erroneous, but simply an undesired part of the signal.

**Line 80:** We wish to maintain the transport reference, as often much of this noise is produced through stochastic processes. We have adjusted the sentence, though, to read "sediment transport and surface perturbations," as the deviations to the topographic surface are what constitutes the noise.

**Line 157:** Added citations for Foufoula-Georgiou and Kumar (1994) and Lashermes et al. (2007) at the end of the next sentence. In addition, the edits we've made to this section no longer have this sentence in its original form, which more clearly necessitated discussing derivatives and convolutions; however, we have added text that discusses why the convolution is useful.

**Line 157:** Convention for the Ricker/Mexican Hat wavelet. It has introduced confusion for us several times, but the Mexican Hat wavelet is explicitly defined as the negative second derivative. We also discussed with the reviewer in person that the output coefficients use a different curvature convention than is typical in geomorphology (convex vs concave values).

**Eqs. 7 & 8:** We have adjusted the wording here (we felt that saying that *s* had no physical meaning to be a bit more abstract than we wished and didn't convey what we were trying to get across). Since our workflow involves selecting  $\lambda$  and solving for s (not the opposite as presented in the review) we considered instead having these equations written as *s*=..., but given we are discussing smoothing scales (corresponding to  $\lambda$ ), we decided to maintain the equations in their current form. That said, we've adjusted more wording here to try and make clear that we are solving for s after selecting a range of  $\lambda$ . We also followed the suggestion of writing the Gaussian and wavelet functions with *s* included. As we elaborated on above, we maintain usage of Torrence and Compo.

**Section 3.3:** The calculation was elaborated on in section 3.1 (i.e. the convolution). We can see how this title introduces confusion, so we've adjusted the section name to "Hilltop extraction."

**Figure 3:** We agree. We had gone back and forth on this, since, as you say, a and b are dependent on our setup. We've replaced a and b with c and d.

**Figure 4:** Perhaps not, but we think some readers will be interested where curvature is measurements are most likely to deviate from each other (i.e. high magnitudes, which partially plays into our synthetic analysis later in the paper). We've opted to keep this here.

**Section 4.1:** This is a great question! We have added an additional subplot to Figure 3 (after removing the extraneous subplots per a previous comment) that shows that the CWT increases in relative speed as DEM size grows. For scaling to large areas, this is a key advantage of the CWT, so thank you again for suggesting this! We have modified language throughout the manuscript that highlights this updated result.

**Line 290:** We refer back to our earlier explanations. They are indeed "different," but it is due to a different way of defining the scale. Rather than thinking about a single value *s* producing different  $\lambda$ , we are picking a single  $\lambda$ , which produces two different *s* (as we mentioned above, we have adjusted text to try and make this clearer). That does, in a manner of speaking, produce two different wavelets, but it is simply the size that differs (the functional form is the same; they are just stretched differently).

**Figure 5C:** These curvature values originally caused some concern, as the hillslopes are so broad and grade so gradually into gentle valleys, that the landscape is largely semi-planar. The mode values near-zero are due to clipping off the positive curvatures from the dataset. While not numerous in number (i.e. number of nodes), the positive curvatures, combined with very low magnitude negative curvatures, results in a mode near 0. Applying  $C_{HT}$  to these hillstops kind of pushes  $C_{HT}$  to its limit, as these hillstopes exhibit such little relief. Fortunately, the mean and median curvatures are negative and produce erosion rates consistent with the CRN erosion rate.

**Figure 6: Thank you!**

**Line 328:** True, though even Roth et al.'s rough site was still comparatively smooth relative to other locations in the OCR. We've added some additional citations. The goal here, regardless, is not to fully mimic a specific landscape, but pick a physically reasonable roughness value. Figure 9 additionally includes the hillslopes with no noise, so the choice of highlighting the 50 cm noisy surfaces seems reasonable.

Figure 7: Suggested change made.

**Line 410:** We haven't checked this specifically with the wavelets. But given the consistency of results between the wavelet and polynomial fits, the results from Grieve et al. (2016) are relevant here. We also refer back to our comment for Lines 73-85. Curvature will be systematically underestimated when using 10-m data. In addition, using a 10-m DEM does not simply constitute taking a high-resolution lidar DEM and using those data spaced every 10 meters. The decrease in data quality and sources with the 10-m DEM is nontrivial.

**Line 427:** We've re-worded the sentence following the suggestion. We've maintained the reference to linear diffusion as a parenthetical comment at the end of the sentence, as we like the reference to linear diffusion in the next sentence.

**Line 530:** We aren't fully sure what the question is. Is it that we used plural "relationships?" (The relationships will depend on diffusivity (i.e. producing multiple different slopes), so we

wish to maintain the plural form). Is it about why we used power law and square root? (We, and Gabet et al., do not always observe a perfect square root form, so we want to maintain the flexibility for a more generic power law, while highlighting the potential square root relationship that Gabet et al. note.).

**Reviewer 2:**

**General Comments:**

**One or two figures of DEMs:**

We have added a figure that has a 3D perspective of our three catchments, with a general focus on each representative hilltop. For each of these 3D-viewed DEMs, we have included a hillslope profile for the representative hilltop.

**Hilltop identification and cutoff parameters:** Please see our response to the comment at line 199).

**Other types of curvature:** Wavelets have been used in geomorphology for other uses, including mapping channel heads and drainage networks. We had alluded to this in the text, but we've added a bit more in the second paragraph of section 3.1. We've also added some more supporting text and some other useful citations in the Discussion. (The Ricker wavelet is specific to the Laplacian, but that is not to say that other wavelets may have the capacity to calculate other types of curvature. That is beyond the scope of this work, however.).

**Figure quality:** We apologize for the rasterization effects for some of the figures. We think this may have been an issue when converting the manuscript file to a pdf. We will try to ensure that the revised manuscript pdf does not have the same problem (as the original figures do not have this issue).

**Guidelines for CHT bias:** We aren't completely sure what is being suggested in regards to making changes to Figure 10b. That figure is already showing the ratio of measured/actual CHT as a function of erosion rate (for a given  $\lambda$ ). Perhaps you are suggesting we include a hillshade of sorts that demonstrates this? Unfortunately, generalization in that way is a challenge for figure 10B, since any hillslope form will depend on the diffusivity (indeed a key point of Figure 10B). But we have added profiles of three example synthetic hillslopes in figure 10A, since diffusivity is limited to a single value.

**Specific comments:**

**Line 73:** We struggled with finding an appropriate acronym for "polynomial functions fit to the topographic surface." We feel that PFT is a reasonably concise acronym (Polynomial Fit to Topography). We have added an additional parenthetical to make this hopefully more obvious.

Line 82: Rectangular window; no. We have modified language to hopefully clarify this.

**Line 88:** This is true. That said, our results demonstrate that the difference in computation speed for the wavelet and polynomial is substantial enough such that if we were to also just apply the wavelets to the hilltops, and not the whole DEM, we would see a similar reduction in processing time. We have chosen not to change the text in this location, as the PFT is slower than the CWT; see our response to the comment on line 406, however, where we did make an adjustment to address this point.

Line 144: The data source is listed in the Code and Data Availability Section. We have added these other details to the text.

Lines 164-165: Adjusted text to clarify the distinction.

**Line 166:** We are not sure what the confusion is referring to here. At this point of the paper, we are discussing the smoothing scale needed to remove topographic noise. Is the confusion surrounding "estimating" erosion rate? We go into how erosion rate is calculated in equation 1. We use the word estimate as it is dependent on D, which we point out later in the paper (and which Gabet et al. emphasize) may exhibit uncertainty. Furthermore, we "estimate" erosion rate, since "measuring" erosion rate is preferably done with cosmogenic nuclides. Unfortunately, broad application of CRNs in expansive regions is often cost prohibitive. Note that for the comment at section 3.4.1 we adjusted language to more explicitly show that we calculated curvature and used hilltop masks to extract  $C_{HT}$  (We think this is the primary source of confusion). Later in the section, we go into details on erosion rate.

Line 192: Added 2.60 GHz CPU

Line 194: Yes. Text adjusted.

**Line 199:** Added statement about restriction of first-order divide lengths (yes, we used the DIVIDEobj function introduced in Schwanghart and Sherler, 2020).

Line 205: We've adjusted this sentence to clarify that we removed drainage divides that share a divide with neighboring drainage basins, where disequilibrium (and thus asymmetry) may be an issue.

Line 210: Added these details.

Section 3.4.1. We adjusted the text to clarify that we measure curvature, then use hilltop masks to extract  $C_{HT}$ . We now wait until later in the section to explicitly mention estimating erosion rate.

The identification of scaling breaks is a specific step to estimate the erosion rate. We've maintained the language we had in this case.

**Line 220:** The D we used here is from previously published work. We added a brief statement to make clear it was estimated in the OCR and added a citation for Roering et al. (1999), in addition to the citation we already had for Roering et al. (2007). In other words, we did not use our hilltop curvature measurements with CRN erosion rates to estimate D, and then use that D to recalculate erosion rate (there's no circularity here).

Line 265: That is simply the non-scaled noise. We added a reference to the  $\pm 1$  m in the sentence that specifies that we are testing amplitudes by scaling those original noisy surfaces.

Line 320: Yes it is, thank you for bringing this to our attention! We have added a sentence at the end of this paragraph pointing that out.

Line 406: At the end of these few sentences, we added a brief statement clarifying this. We stress, though, that the computation advantages of the wavelet would allow for curvature calculation of entire DEMs, with no extra cost! This is of use to not just hilltop enthusiasts, but to those who may wish to calculate curvature elsewhere (i.e. Lashermes et al., 2007; Passalacqua et al., 2010). In addition, while addressing comments from the other reviewer, we made an addition to Figure 3 that shows that the CWT becomes even more efficient as DEM size increase.

**Line 412:** Indeed. While rivers have the advantage that you point out (you don't necessarily need high resolution data for large regional studies), hillslope analyses do not have that luxury. We have opted to keep this sentence in its current form.

**Line 421:** We think that discussing the introduction of planar side slopes into the measurement as hilltops narrow is a fairly conceptual explanation. As you rightly point out, this is fairly easy to visualize for the PFT. For the CWT, however, it is much less intuitive, as the actual calculation takes places in the Fourier domain. Spectral techniques struggle with sharp edges and abrupt transitions (such as sharp hilltops!). Hence, we think that the explanation we've included where "the CWT and PFT kernels have become sufficiently large to incorporate planar side slopes" is a straightforward explanation for why both methods begin to underestimate curvature.

Line 436: Thinking about other filtering schemes is a good point and something we've considered quite a bit. Unfortunately, due to a particular property of convolutions (i.e. taking the derivative of the smoothing function and then convolving it with topography is identical to smoothing to topography and then taking the derivative), your suggestion is what we did here. In other words, the wavelet simultaneously applies a low pass filter (since it's the second derivative of a Gaussian) while calculating curvature. That's what makes this such an intriguing problem; where topographic noise and hilltops have similar characteristic scales, there isn't a clear way to adequately remove the noise while maintaining the underlying hillslope signal. Definitely worthy of future work! We have made some additions to the methods section that makes this property of convolutions clear.

**Figure 9:** We have adjusted the y-axes in the middle two rows, as this was where the most avoidable inconsistencies were. Unfortunately, to make sure the figures are readable, we had to leave the first row ( $E^*=1$ ) figures unchanged. We have added text to the caption, though, that points out that the y-axes between figures in this row differ.

Apologies again on the rasterization. Will make sure the pdf conversion doesn't produce similar issues this time.

**Table 3:** We moved the new sample to its own table and added such details.

---

## Referee Report (RR1)

Review for Struble and Roering, 2021, *Hilltop curvature as a proxy for erosion rates: wavelets enable rapid rapid computation and reveal systematic underestimation*

This is a review of a revised manuscript that I previously offered comments on. In general, I think that the authors have addressed my comments. I briefly reiterate one of my points though.

(1) I remain convinced that the calculation of curvature of a function using wavelets involves the *positive* second derivative of a Gaussian not the *negative* as defined by the Ricker Wavelet. Equation 8 in the revised manuscript and in *Lashermes et al., 2008* demonstrate that derivatives do not involve negative signs of the kernel. I understand that the Ricker Wavelet is formally defined as the negative of the $2^{nd}$ DoG, but it inverts the sign.

This is the only substantial comment that I have that I thought warranted reiterating. I think this is a nice, clear presentation and good work.

---

## Author Response (AR2)

Revisions made to Struble and Roering based on comments by Simon Mudd. *All line number references refer to numbering in original manuscript (as referenced by reviewer).*

**AE, Mudd:**

**Overarching comments:**

$C_{HT}$ **isn't that bad at E\*=10:** For D=0.003 m$^2$ yr$^{-1}$, perhaps not. But for lower diffusivities, it becomes more problematic. We've made adjustments to Figure 11A to emphasize this. We'd also refer you back to earlier changes we made to Figure 11B which demonstrate this too.

**Color Figure 11A by E\*:** Unfortunately, that would simply result in a linear increase of E\* parallel to erosion rate on the x axis. Plus, it would corner us even more into discussing a single diffusivity. Instead, we've opted to expand Figure 11A to include more than a single diffusivity to demonstrate that the deviation between expected and measured $C_{HT}$ can occur at slower erosion rates and depends strongly on D.

**Figure of $C_{HT}$/known $C_{HT}$ as function of E\*:** That is basically what Figure 11A is now. It now allows us to more clearly interrogate D, as opposed to it being hidden within E\*.

**Sentence with major disagreement:** We acknowledge that "likely" is a bit strong, so we've changed it to "may be." Again, though, D is hugely important. If we use a smaller value, closer to D=0.001 m$^2$ yr$^{-1}$, we end up seeing deviation between known and measured $C_{HT}$ >10 percent at erosion rates of 0.04 mm yr$^{-1}$. Does that mean all of the measurements in Gabet et al. are underestimated? Absolutely not. But if you don't know the diffusivity *a priori*, then it is much harder to know how drastically $C_{HT}$ is underestimated from measurements of $C_{HT}$ and E. Unfortunately, for moving forward, this becomes circular ("you need to know diffusivity to accurately measure $C_{HT}$ (or at least know which erosion rates are dangerous for underestimation) to estimate diffusivity"), but that's where we're at.

**"I think what is supported by your paper is that there is a high risk of data artifacts at E\*>10, and that could lead to a spurious E vs D relationship with an exponent less than 1. But I think your paper also shows that, given the erosion rates of sites in our paper, the square root relationship is \*unlikely\* to result from data artifacts alone. Do you disagree with this conclusion?"**

Lots to unpack here. First off, we need to be clear here about what "data artifacts" are. We aren't simply discussing issues with grid spacing and data resolution. This is a geometric problem with the inherent shape of hillslopes. That is, if hillslopes have any kind of planarity near the hilltop, mathematical functions are unable to extract an accurate $C_{HT}$ while *simultaneously* accounting for topographic roughness (both real landscape roughness and that introduced from *actual data artifacts*.). We think you know this, but it is a crucial point. We've made a few minor adjustments throughout the paper to hopefully above confusion about whether artifacts are of geometric origin or from data resolution.

To address your actual question though: per our previous comment, without knowing the diffusivity at the sites in Gabet et al. *a priori*, we really can't answer that question. It may be that the erosion rates at those sites are slow enough and diffusivity is sufficiently high to make the square root relationship unlikely to be spurious. It may also be, however, that diffusivity is lower than you think at some (doesn't have to be all) sites, resulting in an underestimation of $C_{HT}$,

which then leads to an underestimation in D. So, we sadly can't answer your question, as our data only emphasize that D is important (I think we can all agree on that!).

**Our reanalysis of your sites:** Again, we agree that "likely" is strong and premature given our data, but the importance of D that our data demonstrate reinforces that the square root relationship, from our perspective, is not a clear observation clearly free from bias by planar hillslopes. Hopefully, though, this will just motivate future studies!

**Line 51:** Removed redundant phrase, "a nonlinear formulation implied…"

**Line 175:** Yes, that is technically correct. We have reordered material in this section to first introduce this property of convolutions, and then dive into the Ricker wavelet. This way, we have provided a general framework for utilizing the wavelet transform, and then we can discuss the specifics of DoG wavelets. It also avoids most of the confusion around the second derivatives, as we now discuss the convolution property as simply for taking derivatives and not just for calculating curvature.

**Line 184:** Yeah, that's right. See response to previous comment. Isn't that cool, though?!

**Line 189:** Yes, $\lambda$ has dimensions (and so does $s$, since it is the standard deviation of the Gaussian (for the zeroeth order DoG)). To be honest, we've rarely (if ever) seen any papers report units of a wavelet basis function. This may be because most papers utilize wavelets in 1D on temporal data and consider them strictly in the frequency domain. That said, yes, if you follow the units, $\psi$ should have units of $1/L^4$. Throughout the paper, we've added unit designation for each variable where appropriate.

**Additional change at Lines ~185-190:** To address Tyler Doane's continued concern about the way we are defining the two different $\lambda$, we have added some detail on how both Lashermes et al. (2007) and Torrence and Compo (1998) define $\lambda$. We've discussed this with Tyler, and our understanding of his concern is that because $\lambda$ is different for the two definitions, then the wavelets are different and, therefore, the outputs aren't comparable. Granted, the wavelets are "different," but only in as much as two parabolas with a different coefficient are different. The functional form is the exact same; we simply use a different definition for the scale, which "stretches" the wavelet and considers topography with a different characteristic wavenumber. The outputs are certainly comparable, since we are calculating curvature for both. Otherwise, we wouldn't be able to compare to the output from the 2D polynomial either. We have added brief statements that clarify the methods by which Lashermes et al. and Torrence and Compo define $\lambda$. We personally find these definitions to be a bit superfluous and unnecessary to understand that $\lambda$ dictates the smoothing scale of the wavelet (and we fear this new added explanation may confuse this understanding for some), but we acknowledge that some readers may appreciate this specificity.

**Line 195:** We do not use the wavelet to define the smoothing scale for the polynomial. We independently define the values of $\lambda$ ahead of time, with the only stipulations being that s>1 for the wavelet (since the convolution can't be applied at scales smaller than the grid spacing; translates to $\lambda$>5 for 1-meter data) and that $\lambda$ be odd to accommodate the window-size (diameter) requirement for the polynomial. Thus, we applied both the CWT and PFT for odd $\lambda$>5 (at irregular intervals for computations efficiency at larger $\lambda$; see Figure 3). We've added some text back at the start of section 3.1 to hopefully clarify that the PFT is applied for a range of $\lambda$

independently from the CWT. (We also clarify that λ for the PFT is the diameter of the smoothing window).

**Lines 232-234:** We adjusted language here to avoid confusion about hillslope length vs the "length" of the hilltops (how far *along* the hilltop we measure $C_{HT}$).

**Line 239:** Correct. We now specify this (that we use diameter here) more clearly in section 3.1. But you are correct in that Roering et al. report the value as a radius of 7.5 m.

**Line 275:** Added some sentences that give a bit more detail.

**Lines 399-403:** Thank you!

**Line 474:** Removed modest; though, we maintain the rest of that sentence.

**Line 494:** Unfortunately, no. It all depends on the curvature of the noise signal. For instance, the overestimation of curvature at modest smoothing scales (λ≈7-15m) in some cases (Figure 10C, D, G, H, I) corresponds to locations where negative curvature with a higher magnitude than the underlying hillslope form biases the value to more negative values (some underestimations may be greater than they would be without noise due to a similar issue). An early version of the manuscript discussed this more, but we felt it distracted greatly from the more notable underestimation seen at most smoothing scales, and importantly, when no noise was added. Overall, this is more of a question regarding topographic noise, which for our purposes here, is poorly constrained and is the subject of future papers. We've added a statement to this effect.

**Line 567:** Figure 11A is diffusivity dependent, which is why we originally added Figure 11B. If the diffusivity is lower, then the erosion rate at which this deviation occurs is also lower. So, to help make this clearer and to avoid misinterpretation of Figure 11A as being the absolute erosion rate at which this problem manifests, we've adjusted Figure 11A to also include a range of diffusivities (0.001-0.003 $m^2$ $yr^{-1}$). This is certainly not an exhaustive examination of the range of diffusivities found in natural landscapes. But it demonstrates that if you visit a landscape where diffusivity is low, then you are *more likely* to underestimate $C_{HT}$ at slower erosion rates. We acknowledge that given more work is needed to clarify this problem, our language is perhaps a bit strong. We've modified "likely" to "may be." We think this is very reasonable given the analysis we've completed.

**Fig. 1:** Added this statement.

**Fig. 2:** Font size adjusted.

**Fig. 3:** Adjusted font size (in this figure and others). Added L07 and TC98 definitions.

**Fig. 5:** Added L07 and TC98 to appropriate figures.

**Fig. 11:** Since we chose a range of E*, then yes, hillslope length is relevant. But if we want to utilize Equation 1 ($E = -\frac{\rho_s}{\rho_r} D C_{HT}$), then we want to isolate the $C_{HT}$ term. The hillslopes are 100 m long. We have added that information in the same parenthetical where we specify the range of diffusivities.